# Non-invasive eye tracking and retinal view reconstruction in free swimming schooling fish
Ruiheng Wu[1,2], Oliver Deussen [1,2], Iain D. Couzin [2,3,4] & Liang Li [2,3,4] ✉

Eye tracking has emerged as a key method for understanding how animals process visual information, identifying crucial elements of perception and attention. Traditional fish eye tracking often alters animal behavior due to invasive techniques, while non-invasive methods are limited to either 2D tracking or restricting animals after training. Our study introduces a non-invasive technique for tracking and reconstructing the retinal view of free-swimming fish in a large 3D arena without behavioral training. Using 3D fish bodymeshes reconstructed by DeepShapeKit, our method integrates multiple camera angles, deep learning for 3D fish posture reconstruction, perspective transformation, and eye tracking. We evaluated our approach using data from two fish swimming in a flow tank, captured from two perpendicular viewpoints, and validated its accuracy using human-labeled and synthesized ground truth data. Our analysis of eye movements and retinal view reconstruction within leader-follower schooling behavior reveals that fish exhibit negatively synchronised eye movements and focus on neighbors centered in the retinal view. These findings are consistent with previous studies on schooling fish, providing a further, indirect, validation of our method. Our approach offers new insights into animal attention in naturalistic settings and potentially has broader implications for studying collective behavior and advancing swarm robotics.

Understanding how animals in groups obtain information about social partners, as well as non-social (e.g., environmental) information, is essential for unraveling the core mechanisms underlying collective behavior[1,2]. Given that many social species rely on vision to acquire information[3] and make behavioral actions accordingly, it is crucial to reconstruct the visual input of individuals within a social context to better understand the dynamics and processes of information transfer. For instance, by recreating retinal views and correlating them with movement decisions, we can uncover the visual-motor mechanisms underlying visually-mediated collective behavior. Previous studies have attempted to reconstruct the angular area through technical methods such as ray casting over fish eyes in 2D to estimate visual information transfer[4,5], yet these studies often overlooked the impact of eye movements.

In nature, eyes are constantly in motion, providing information about the environment[6,7], prey localization[8], and salient cues to inform social interactions[9]. Numerous studies have correlated gaze targets with social perception, seeking evidence of vision-based social interactions across various species. For example, mice combine head and eye movements to survey their surroundings and participate in social interactions while visually following objects[9]. Common marmosets move their eyes to scan the face region to recognize conspecifics[10]. In birds, such as European starlings, individuals perform lateral scans to gather surrounding information and social cues simultaneously[11]. Goldfish, in particular, exhibit complex and controlled eye movements, characterized by regular saccadic steps and spontaneous side-to-side movements, which are crucial for stabilizing their visual field, gathering environmental information, and engaging in the acquisition of information employed in social learning[12–14].

Although eye-tracking has long been developed for humans and animals[15–17], non-invasive eye tracking of freely moving animals in large space in 3D is still challenging. Most previous studies of eye tracking in freely swimming fish have been limited to 2D environments[18–21]. For example, a simple blob detection method can be effectively employed for eye tracking of larval zebrafish due to their transparent body, distinct black eyes, and limited 3D swimming capabilities[18,19]. However, this does not work effectively for free swimming adult fish in groups, where eyes could not be easily

[1]Department of Computer and Information Science, University of Konstanz, 78464 Konstanz, Germany. [2]Centre for the Advanced Study of Collective Behaviour, University of Konstanz, 78464 Konstanz, Germany. [3]Department of Collective Behaviour, Max Planck Institute of Animal Behavior, 78464 Konstanz, Germany. [4]Department of Biology, University of Konstanz, 78464 Konstanz, Germany. ✉e-mail: lli@ab.mpg.de

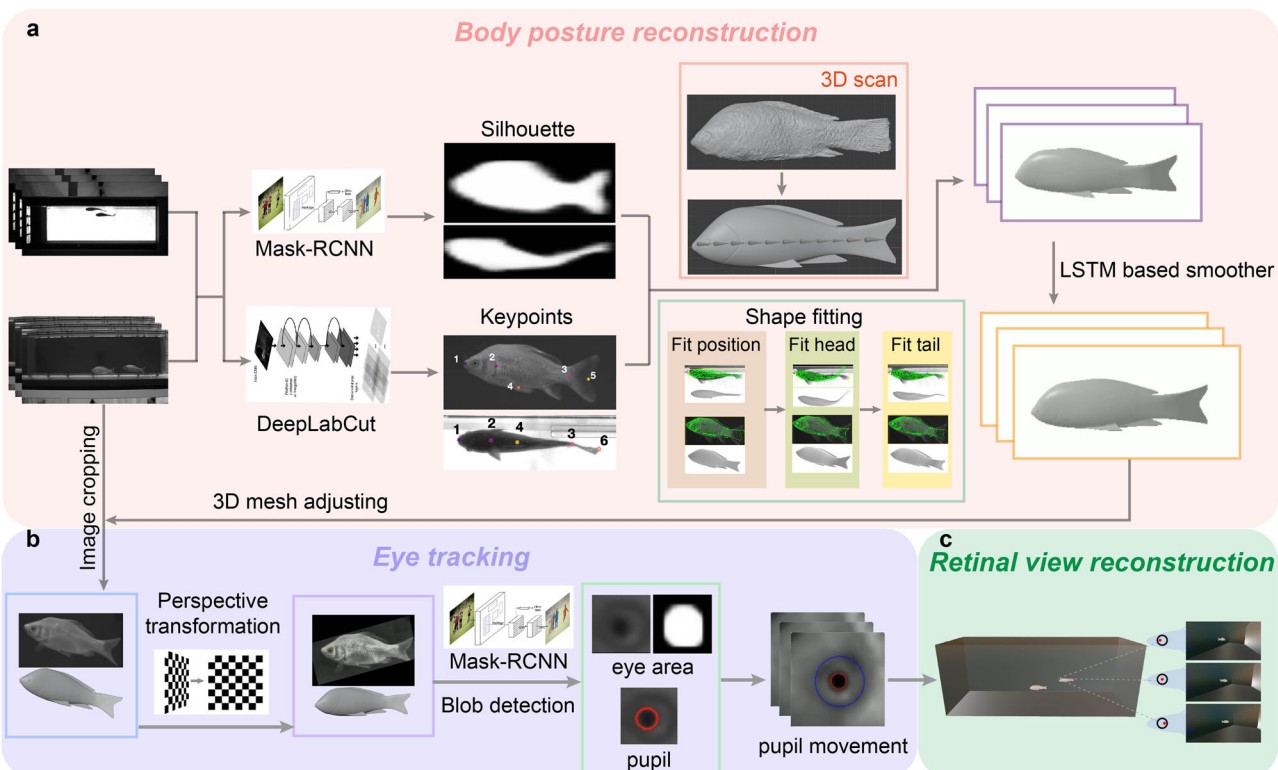

**Fig. 1 | Schematic of eye tracking and retinal view reconstruction of freely swimming schooling fish in 3D.** Three main modules are included in the task: 3D posture reconstruction based on DeepShapKit[24] (**a**), eye tracking (**b**), and retinal view reconstruction (**c**).

distinguished from their body textures, and eyes move in 3D rather than 2D. Recording from the primary visual cortex (V1) using a multi-electrode array allows to provide high-resolution eye positions[22]. However, this is an invasive method and is not easily applicable to free-swimming fish in 3D environments[22]. With small plastic markers on the fish's eyes, Ben-Simon et al.[8] successfully obtained 3D eye movement data for a single free swimming fish. However, the plastic marker could potentially alter the animal's behavior too. On the other hand, deep learning methods, employing images from multiple-view cameras or even a single-view camera, nowadays have the potential to track 3D eye movements directly[23]. However, their practical application is currently limited due to the shortage of large, precisely labeled datasets required for training.

In this paper, we present a novel, non-invasive, and restraint-free method for tracking the eye movements of freely swimming fish in groups in 3D. Our technique uniquely determines the 3D gaze direction of each eye (as long as it is visible in recording) on free-swimming schooling fish in a relatively large space without the need for markers on either the body or eyes, relying exclusively on images captured by cameras. In general, our eye tracking process has three main modules: 1) 3D body posture reconstruction, 2) eye tracking, and 3) retinal view reconstruction (Fig. 1, Supplemental video 1). For the first module, we obtain 3D posture by reconstructing the body mesh with DeepShapKit[24], which integrates silhouettes and key points of the fish body extracted by deep learning algorithms (Fig. 1a). For the second module, we conduct a perspective transformation based on the 3D posture and track the eye position and movement, taking into account the posture of the fish, through blob detection of the pupil in the cropped eye area of the images (Fig. 1b). For the final module, the retinal view of neighboring fish is reconstructed using the 3D mesh bodies of both fish and the eye movement data of the focal fish (Fig. 1c). We validated our methodology with synchronized videos from two perpendicular views of two goldfish swimming freely in a flow tank at various flow speeds ranging from 1.2 BL/s (Body length per second) to 1.6 BL/s with an interval of 0.1 BL/s (see

supplementary Fig. 1 and[25] for details). A comparison of our eye tracking with human-labeled, as well as synthesized ground truth data validates our approach. Indirect verification was performed by tracking the eye movements of one fish following another. We observed that the right eye's movement was positively correlated with the position angle of the leader on the right side but negatively correlated when the leader was on the left side. The observed eye movements are consistent with findings from previous studies[13,26]. Finally, the reconstruction of the retinal view revealed that fish tend to keep the leader centered in their retina on the side of the leader fish while following, indirectly supporting the effectiveness of our eye-tracking method as well.

## Results

### 3D body posture reconstruction
To accurately track the eyes of freely swimming fish in 3D, our initial step involves reconstructing the body posture of the fish in 3D. We applied DeepShapeKit[24], the process is summarized in Fig. 1. We begin by 3D scanning the fish to create a benchmark 3D model. Next, we optimize the position and kinematics of this model to minimize the discrepancies between the model and the silhouettes of real fish tracked by Mask-RCNN as well as key points of the body central line tracked by DeepLabCut. An LSTM (long short-term memory network)-based[27] smoother is applied to smooth the 3D mashes in sequence. With the 3D body mesh over the global coordinate, we get fish body posture in 3D.

### 3D eye tracking
We mainly involve two main steps to track the continuous movements of fish eyes in 3D: perspective transformation and eye detection (Fig. 1). Since fish swim in 3D, exhibiting motions characterized by rolls, yaws and pitches, their eyes cannot be consistently captured in a perpendicular view by a fixed camera, making perspective transformation necessary. To address this issue, we employ a network called Mask-RCNN[28] to isolate how each fish appears in each view and utilize a perspective transformation to convert the 2D

cropped image into a 3D representation based on the orientation of the fish's meshed body:

$$C_{x,y} = H \times I_{x,y} \qquad (1)$$

Where, $H$ is the transformation matrix; $I_{x,y}$ represents the original image, where the fish's body is tilted in 3D space and projected on the 2D image; $C_{x,y}$ is the pixel coordinate on the transformed image. The transformation matrix $H$ ($3 \times 3$) is calculated based on the correspondence between the fish's 3D pose and the target pose (fish body parallel to the image plane), as shown in detail in the method section and supplementary materials. We then applied the Mask-RCNN again to detect the eye sclera (large light disk) with only 50 hand-labelled images for training. We then applied threshold-based blob detection for the pupil (small dark disk) on the transformed image. The eye sclera and pupil positions are then transformed back to the original perspective in 3D as the position of the eye of real fish in 3D (see methods).

### Reconstruction of retinal view

We applied Blender[29], an open-source 3D modeling and animation software, to reconstruct the group swimming dynamics of fish in a tank and to estimate the retina view from the perspective of a following fish's eye using virtual cameras (Fig. 2).

The fish models are based on the 3D mesh body tracked with both kinematics and movements for each fish (Fig. 2). We then place each fish model with corresponding kinematics at the corresponded positions in 3D for each frame (as described in the method section). Subsequently, we set a virtual camera which is controlled by the eye movement (in both position and rotation) and estimate the retina view from the fish's eye (see method for details). We set the virtual camera's field of view to 190 degrees, mimicking that of a goldfish's eye[30], resulting in a 50° blind spot behind with eyes.

### 3D eye tracking verification

We evaluated our eye-tracking method using two approaches: using manually labeled data after the perspective transformation, and using synthesized side-view images with randomly initialized pupil sizes and positions on the sclera. The human-labeled data and synthesized images served as ground truth after and before perspective transformation, respectively.

**Labeled fish eye images.** We sampled 110 fish images from the videos and transformed them according to the fish's pose. We then marked the centers and sizes of the sclera and pupil on the fish images using circular markers. After that, we compared the detected positions and sizes of the pupil and sclera with the manually labeled ones (Fig. 3). As shown in Supplementary Tab. 2, we observed a view angle error distribution with a mean of $-2.445$ degrees and a standard deviation of 2.650 degrees, within a range of $0 \pm 9$ degrees. The error in pupil position lies within $0 \pm 0.5$ mm, with a mean of $-0.072$ mm and a standard deviation of 0.120 mm. For the sclera position, the error lies within $0 \pm 0.5$ mm, with a mean of $-0.041$ mm and a standard deviation of 0.116 mm.

**Synthesized fish eye image.** To validate our eye-tracking method before perspective transformation, we synthesized side-view images with generated ground truth for eye position. First, we filled a square with dimensions w × w using the average color value from the area surrounding the fish's eye in the original side-view image. At the center of this square, a light circle, representing the eye sclera, is drawn with a diameter of $\frac{2w}{3}$. For the artificial pupil, a dark circle is placed within the light circle. The size and position of this pupil are randomly determined, adhering to a uniform distribution. The radius varies within a range of $[\frac{w}{6} - \frac{w}{20}, \frac{w}{6} + \frac{w}{20}]$ pixels, while its position is defined to be within $[\frac{w}{2} - \frac{w}{6}, \frac{w}{2} + \frac{w}{6}]$ pixels. We chose this range to ensure that the distance between the pupil and sclera borders is minimal, replicating the

**Fig. 2 | Illustration of the three steps involved in retinal view reconstruction based on eye movement tracking.** First step (yellow): position a camera at the tracked eye position in 3D. Second step (green): rotate the camera to align with the fish's view direction. Final step (pink): configure the camera parameters based on the structure of the fish's eye.

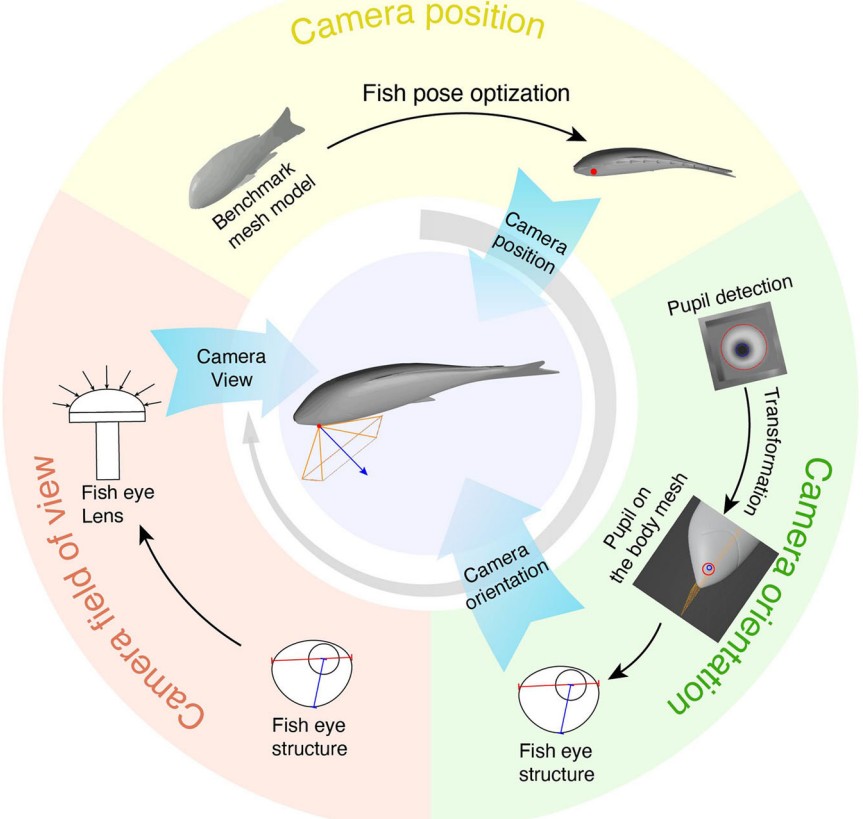

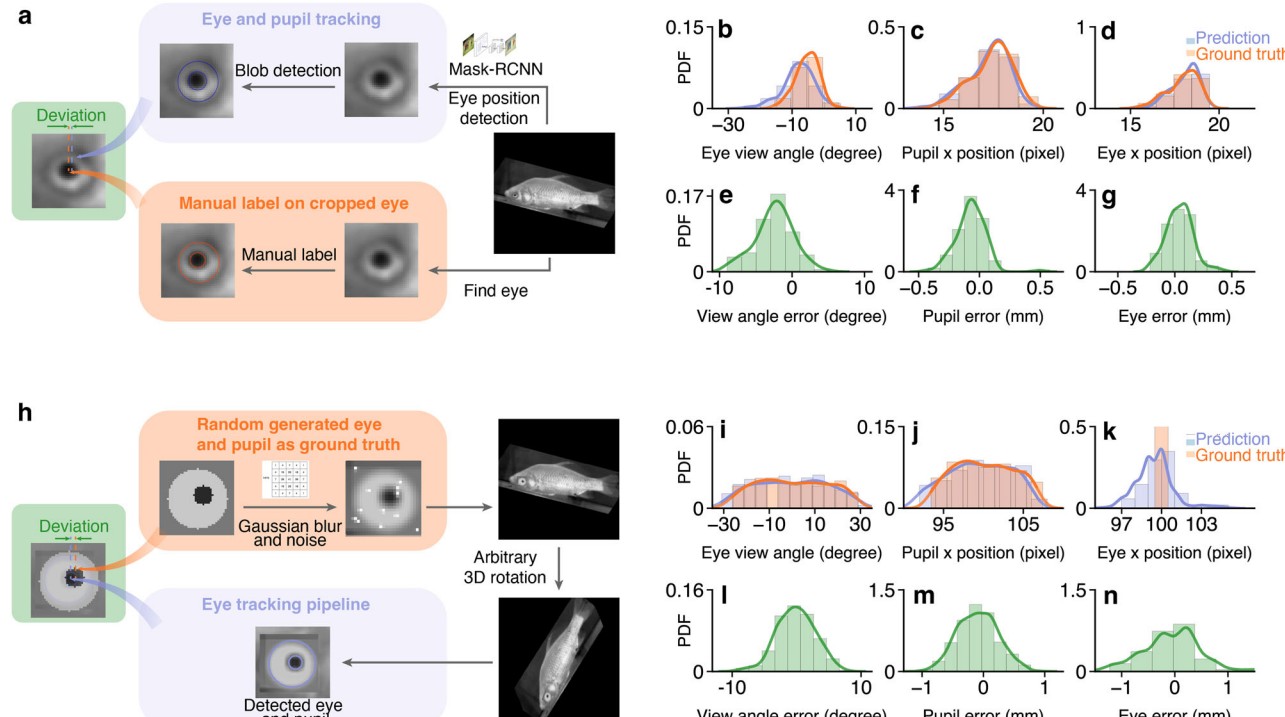

**Fig. 3 | Evaluation of the eye tracking. a** The pipeline of manually labeling eye positions after perspective transformation and conducting eye tracking as a comparison. Probability distribution function (PDF) of **b–d**: ground truth and detected eye view angle, pupil position, and eye position. **e–g**: deviations between the ground truth and eye detection outputs. **h** The pipeline of generating synthesized data as ground truth and conducting eye tracking for a comparison. The PDF of **i–k** ground truth and detected eye view angle, pupil position, and eye position (**k**). **l–n** deviations between the ground truth and eye detection outputs. The deviations are scaled based on the average eye diameter (6.75 mm).

appearance of the eye when the fish turns its head toward the camera. Next, we perform a perspective transformation to tilt the images, introducing a distortion that aligns with the fish's body as captured in the camera's view. Finally, we apply a Gaussian blur to the eye with a kernel size of $\frac{w}{6}$, add white dot noise at random locations amounting to 0.05% of the total pixel count, and place the synthesized eye image onto the fish's eye location in the video, as illustrated in Fig. 3h.

We subsequently applied our eye-tracking algorithm to track the eye movements of fish within this synthesized dataset. Results with 500 synthesized images are shown in Fig. 3 and Supplementary Tab. 2. In general, we observe an view angle error within $0 \pm 10$ degrees, with a mean of $-0.099$ degrees and a standard deviation of 2.950 degrees. For the pupil position, the error lies within $0 \pm 1$ mm, with an average error of $-0.090$ mm and a standard deviation of 0.326 mm. For the eye sclera position, the error lies within $0 \pm 2.5$ mm, with an average error of $-0.016$ mm and a standard deviation of 0.628 mm.

**Tracking quality across different view angles**

Since our tracking method uses blob detection for the pupil position, potential issues may arise when the fish's pupil is too close to the edge of the sclera. To assess whether the position of the pupil affects our tracking quality, we analysed the tracking quality across the different positions of pupil. We split the data into two groups: center cases, where the pupil is centrally positioned, and edge cases, where the pupil is positioned near the edges (see Supplementary Fig. 6). Results are summrized in Supplementary Tab. 4. For manually labeled data, the mean view angles were $-2.42°$ for the edge cases and $-2.47°$ for the center cases, with standard deviations of $3.02°$ and $2.23°$, respectively. For synthesized data, the mean view angles were $-0.43°$ for the edge cases and $0.31°$ for the center cases, with standard deviations of $3.16°$ and $2.70°$, respectively.

**Eye movement and retinal view reconstruction in schooling behavior**

To further assess our eye movement tracking and retinal view reconstruction, we applied the algorithms to two fish swimming in a flow tank with two perpendicular views, and then compared the retinal view and eye movement with their schooling behavior. We first applied the following filters to extract leader-follower schooling behavior by: the position angle, defined from the leader's position to the follower's local coordinate, is less than 40 degrees ($|a| < 40°$); the front-back distance is less than 0.4 meters ($|x| < 0.4$); and the distance between the two fish is less than 0.45 meters ($d < 0.45$ meters). As a result, we obtained a dataset comprising 45,499 data points, divided into 7860 cases where the leader was in the front-right position relative to the follower ($a > 0$), and 37,639 cases where the leader was in the front-left position ($a < 0$) (see Fig. 4a). Over the leader-follower schooling behavior, the front fish mostly appears within 100° in front.

We then applied our eye movement tracking to the follower and analyzed the correlation between eye movements and the relative position angle $a$ between the two fish, as shown in Fig. 4b. The eye position was normalized based on the sclera, so that we could compare the amount of eye movement between fish with different eye sizes. We observed that the relative position angle $a$ is positive when the leading fish is on the left side of the following fish's head direction (as illustrated in Fig. 4b), and negative when the leading fish is on the right side (the correlation between $x$ and $a$ is $r = -0.69$). This means that when a fish sees another in front, the position angle between them causes the eye on the same side as the neighbor to move more forward, while the opposite eye may move backward. These eye movements during leader-follower behavior are consistent with previous studies[31,32], indirectly suggesting the effectiveness of our algorithm. Additionally, since there is an asymmetry in data amount between the leader on the left and right of the follower is primarily due to the asymmetric boundaries of the flow tank, we also conducted bootstrap with same

**Fig. 4 | Eye tracking of two swimming goldfish exhibiting leader-follower behavior. a** Selected pairs of goldfish within swimming groups. (i) Two-fish relationship from the bottom view. (ii) Front fish position density map on a 3D sphere. All 45,499 pairs of data are binned in 12 degrees increments, with a radius of 17.5 cm. (iii): The front fish position heat map, viewed from the bottom side, shows the relative positions of the front fish from the perspective of the following fish looking upwards. Each point on the heat map represents the position of the front fish, with the following fish's position fixed at the center. The $x$ and $y$ axes span a range of $\pm 50$ cm, capturing the spatial distribution of the front fish. (iv): The front fish position heat map, viewed from the front side, presents the relative positions of the front fish from the perspective of the following fish facing left. Again, each point represents the front fish's position, with the following fish at the center. The x and y axes cover a range of $\pm 50$ cm, illustrating the positioning patterns during the interaction. **b** The correlation between the normalised eye movements ranging from $-0.4$ (rightmost position) to 0.4 (leftmost position) and relative position angles.

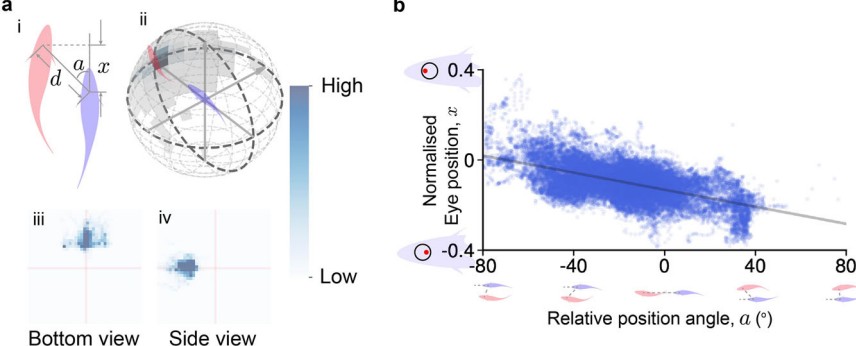

analyses. These analysis results show similar correlation with an average slope of $-0.00195$ and an average intercept of $-0.132$, with standard deviations of $3.46 \times 10^{-5}$ and $6.85 \times 10^{-4}$, respectively (see Supplementary Fig. 7)

Finally, we applied retinal view reconstruction to estimate how the leader appeared from the follower's perspective during the following behavior. For comparison, we reconstructed the retinal views under conditions where the eyes were either stationary or moved randomly (as shown in Fig. 5a). In these control conditions, the leader's position on the follower's retina varied much more significantly (as illustrated in Fig. 5b) compared to when the eyes were moving intentionally, as observed in our experiments (refer to Supplementary Tab. 1). This suggests that fish deliberately adjust their eye movements to maintain their neighbor, in this case, the leader, near the center of their retina. This behavior is also consistent with previous reports[33–35], indirectly verifying our method.

## Discussion

In our study, we introduced a novel method to non-invasively track eye movements of free-swimming fish schools. This method includes 3D body shape reconstruction, perspective transformation of the camera view, and pupil detection and tracking. We validated this approach using an augmented dataset. Additionally, we developed a technique to reconstruct the retinal view of the front neighbor from the perspective of the following individual, which involves scenario reconstruction, adjustment of the camera view's position and orientation, and view reconstruction. We finally validated our eye tracking and view reconstruction with free-swimming goldfish in leader-follower schooling behavior. Results show that eye movements on either side of the fish are decoupled, allowing for focused attention on their neighbor. The results of our retina view reconstructions show that fish dynamically adjust their eyes in an attempt to maintain their neighbor's image at the center of their retina when the attention is locked.

Our methods primarily capitalize on 3D body shape reconstruction. By analyzing body shape data, we accurately ascertain the position and orientation of the eyes. This technique marks a significant advancement over the current predominantly 2D approaches in non-invasive fish eye tracking[4,21,36]. Our analysis of schooling behavior using this method provides results similar to those previously reported (Figs. 4 and 5), indicating that the precision of our eye tracking is sufficient for biologically meaningful insights. The primary requirement of our method is that each fish must be

captured by at least two cameras, with the eyes being recorded by at least one camera. Therefore, while our study mainly showcases a two-camera setup, our method is theoretically scalable to include any number of cameras. The more cameras there are, the higher the precision, especially when multiple views capture eye movements. However, this comes at the cost of training Mask-RCNN and DeepLabCut to extract fish masks and keypoints for each view. The same training process is required when applying our method to other fish species.

Additionally, although our current example features two fish swimming in relatively similar directions, our method is versatile enough to be applied to scenarios with freely-swimming fish in large numbers. Based on detection accuracy, we can identify which fish the tracked fish is looking at if there are two leading fish separated by at least 5° in its field of view. In general, the more individuals involved, the more challenging it becomes due to frequent overlapping. This would require not only more cameras but also a large, highly precise dataset of fish mask and keypoint labeling for training.

In our synthesized validation data, we simulate conditions similar to those observed in our fish recordings, including variations in illumination, shape, slight tilting, and underwater noise such as small particles. However, these simulations may not fully capture all potential noise or variations in eye anatomy that could appear in other video recordings. Through comparison with human-labeled data, we demonstrated that our pipeline is effective for the videos we collected. Nevertheless, we recommend adjusting parameters, such as threshold values in the blob detection step, when adapting our method to different datasets. In cases where the fish is frequently occluded, adding more cameras can help maintain a consistent view of the eye.

Our retinal view reconstruction offers more realistic visual inputs by accounting for eye movements, as opposed to the previous 2D angular area estimation methods[4,18,20,21,36]. In the case of two fish swimming, we observed that when one individual swims in front, the follower tends to lock onto the individual by centering their view in the eye. This approach can be applied to analyze target of gaze in scenarios involving multiple neighbors[33–35], by determining which neighbor is closest to the center of the retinal view. The negatively synchronised eye movement suggests that they might be able to lock onto two different individuals simultaneously with each eye[31,32]. However, since we previously could not track the opposite eye, we can quantitatively measure the eye movements on both sides in the future experiment with additional cameras capturing both eyes.

**Fig. 5 | Application of retinal view reconstruction in two swimming goldfish. a** follower's retinal view of the front individual with tracked eye movements (i), static eye (ii), and randomly moved eye (iii). **b** position of the front individual on the follower's retina along the $x$-axis (i) and $y$-axis (ii), and their corresponding variances (iii and iv) as determined by bootstrap analysis.

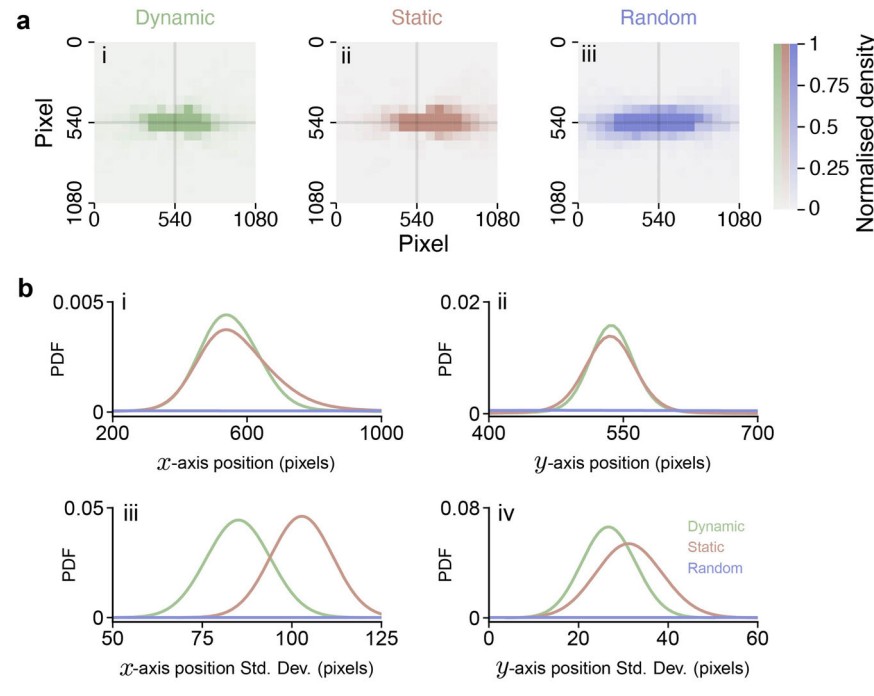

## Methods

### Hardware and data collection

A flow tank, capable of generating laminar flow, was utilized to capture videos of fish swimming. A detailed illustration for this recording setup is presented in Supplementary Fig. 1. The test arena measured 88 cm in length, 25 cm in width, and 25 cm in depth. Videos were recorded using two synchronized cameras capturing bottom and side views at 90 frames per second. In the bottom view recording, backlighting was used, rendering the texture invisible. In the side view, normal light was employed, allowing the texture to be visible. We randomly picked two fish for each experiment from a group of 32 goldfish in our animal care facility. All animal handling and experimental procedures were approved by Regierungspräsidium Freiburg, 35-9185.81/G-17/90.

Regarding the camera information, recordings were made using the Basler acA2040-90umNIR camera model, equipped with a Kowa LMVZ166HC lens, and an original resolution of $2040 \times 2040$ pixels. We crop the video to a resolution of $2040 \times 1040$ pixels, and adjust the focal length to 39.4 millimeters. Calibration of the camera was conducted using a chessboard pattern submerged in water and the MATLAB camera calibration function. Image distortion in the front view was corrected using parameters obtained from the calibration process (Supplementary Fig. 2). The recorded data included some clips that did not provide significant information for the target of gaze study. For instance, when there was a considerable distance along the z-axis (vertical direction from the front view), it became challenging for the fish to see each other when swimming forward. To address this, video data were filtered by selecting frame sequences based on the detected fish positions. In our setup, frames were excluded where the angle between two fish on the x-z plane exceeded 40

degrees and where the distance between two fish on the x-axis was less than 0.2 meters (the size of the fish body). This ensured that the fish swam together at nearly the same height, and the leading fish did not occlude the following fish's head. It's important to note that due to the exclusion of these frames, the videos were divided into several segments. Additionally, we ensured that the filtered sequence comprised at least 40 frames for ease of analysis and visualization.

### Statistical methods

In Fig. 4, we apply bootstrap analysis to determine the regression function that describes the relationship between normalized eye movements and relative position angles. In each iteration, we randomly sample 3000 data points from both the left cases ($x<0$) and the right cases ($x\geq0$), resulting in a total of 6000 points per batch. We calculate the regression function for each batch and repeat this process for 200 iterations.

In Fig. 5, we apply bootstrap analysis to calculate standard deviation of leader fish's position on $x$-axis and $y$-axis. We randomly sample 1000 data points from the result leader fish positions and calculate the standard deviation within the sample for 200 iterations, and plot the 200 standard deviations as a distribution curve. This sampling method is applied to $x$-axis and $y$-axis independently.

### Fish swimming pose reconstruction

In our eye tracking workflow, the 3D pose of the fish plays a crucial role as it allows us to conduct perspective transformation for the eyes and visualize swimming records. The fish's pose is reconstructed from video recordings using the method we published in our previous publication[24]. Our approach involves simultaneous video recording from different angles, and we fit our predefined 3D mesh to each frame to minimize keypoint and silhouette errors.

Our experimental setup includes two view angles: front view and bottom view. We employ the Mask R-CNN network[28] to obtain the fish silhouette (Supplementary Fig. 2) and DeepLabCut[37] to extract the keypoints, as illustrated in Fig. 1, to reconstruct the body pose of fish. We chose the Mask R-CNN network for its strong performance in detecting objects along with their segmentation masks[28], and DeepLabCut for its ability to reliably track keypoints on multiple animals' bodies[38]. We set a skeleton along the fish anteroposterior axis, and deform the fish body with linear

blend skinning (LBS) In this way, we can adjust the fish model's body pose by defining its skeleton's length and rotation. If we stretch one section on the skeleton, which we call it a bone, the corresponding part on the fish model will be longer and the vertex density on that part will reduce. The rotation of all the bones together decide the fish model's body curve. We restrict the bones to have no pitch rotation, a slight roll rotation, and mostly yaw rotation, as a fish swims normally by bending its body left and right[39]. Subsequently, we use the silhouette and keypoint positions on each input frame to refine the body pose.

The pose fitting consists of three parts: (1) global pose fitting, (2) body pose fitting, and (3) tail position fitting. In each step, we focus on different parts of the fish body and let the fish model change to a shape close to the input frame by adjusting the skeleton parameters. The fish model is adjusted in this three step manner instead of tuning all parts together so that we can reduce the possibility of converging to a strange shape.

### Retinal view reconstruction

The orientation of the virtual camera within the 3D scene is dynamically adjusted in each frame based on the detected eye movements, as illustrated in Fig. 2.

The first step is to adjust the camera's position. By default, the camera is positioned on the eye of the fish mesh model, aligned with the normal direction of the model surface. This corresponds to a scenario where the pupil is situated at the center of the eye area. During the adjustment of the mesh model's pose, the camera's position is transformed concurrently with the mesh model, ensuring it consistently remains at the eye position.

Subsequently, the correct camera orientation is determined. As the pupil moves, the camera dynamically rotates to track the direction of the pupil. This is achieved by retrieving the pupil's position from the eye tracking results and applying the reverse transformation, as demonstrated in the eye tracking portion (Supplementary Fig. 4). This reverse transformation unveils the 3D position of the pupil on the mesh model. Following the fish eye structure (Supplementary Fig. 5), the camera is placed at the retina center of the fish's eye, positioned beneath the mesh model surface at a depth that matches the eye size. The view direction of this camera points from the retina to the pupil center based on the reverse transformation. Finally, the camera's view angle is adjusted to 190 degrees to match the fish's field of view. The reconstruction of the fish's vision is achieved by capturing the rendered scene through the output of the virtual camera.

### Reporting summary

Further information on research design is available in the Nature Portfolio Reporting Summary linked to this article.

## Data availability

All data supporting this study's findings have been publicly uploaded on figshare[40] (https://doi.org/10.6084/m9.figshare.25886437).

## Code availability

All the data analyses were performed using custom scripts written in Python (Python Software Foundation, 2018). Our published codes are licensed under CC-BY-4.0. All codes supporting this study's findings have been uploaded on figshare[40] (https://doi.org/10.6084/m9.figshare.25886437) and are publicly available. A description of published data and code is presented in the supplementary information and Supplementary Table 5.

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

## Acknowledgements

We thank all members of the Department of Collective Behavior of Max Planck Institute of Animal Behavior who assisted with the project: We thank the animal care staff at the University of Konstanz, including Alexander Bruttel, Christine Bauer, Jayme Weglarski, and Dominique Leo, for help in taking care and prepare fish for experiments. The project was funded by the Max Planck Institute of Animal Behavior. I.D.C. acknowledges support from the Office of Naval Research (N00014-64019-1-2556), the European Union's Horizon 2020 research and innovation programme under the Marie Skłodowska-Curie grant agreement (860949), the PathFinder European Innovation Council Work Programme (101098722), and the Deutsche Forschungsgemeinschaft Gottfried Wilhelm Leibniz Prize 2022 584/22. All authors acknowledge support from the Deutsche Forschungsgemeinschaft (DFG, German Research Foundation) under Germany's Excellence Strategy-EXC 2117-422037984 (to I.D.C. and O. D.) and the Max Planck Society. L.L. acknowledges funding support from the Max-Planck Society, the Deutsche Forschungsgemeinschaft (DFG, German Research Foundation) under Germany's Excellence Strategy–EXC 2117-422037984, the Sino-German Centre in Beijing for generous funding of the Sino-German mobility grant M-0541, and Messmer Foundation Research Award.

## Author contributions

L.L. and R.W. conceived the idea and designed the project; R.W. conducted the experiments and collected data, R.W. wrote the code of software, L.L. and R.W. conducted the analyses and data validation under the supervision of O.D. and I.D.C.; L.L. and R.W. wrote the initial draft and all authors contributed to the revision of the text.

## Funding

## Competing interests

The authors declare no competing interests.
