## [Transparent Peer Review file · Communications Biology]

Non-invasive eye tracking and retinal view reconstruction in free swimming schooling fish

Corresponding Author: Dr Liang Li

Version 0:

Reviewer comments:

Reviewer #1

(Remarks to the Author)

The authors present a method to quantify eye movements during social interactions of freely swimming adult goldfish in three dimensions. They rely almost entirely on previously published methods, but ingeniously combine them into a pipeline ranging from fish tracking and body pose estimation to eye tracking and the deduction of retinal views. With this composite method, the authors document eye movements during two-fish shoaling/schooling behaviour, in which the follower keeps one of its eyes oriented towards the leader and performs conjugate eye movements including the other. To assess tracking quality, the authors devise a benchmarking procedure based on artificially generated video frames with known eye positions.

The manuscript is method-focused. Making such a method available to the community is very valuable, and I tend to support its publication in *Communications Biology*. But some questions remain regarding benchmarking and re-usability. In particular, the authors should elaborate on their assessment of tracking quality, and on what code will or will not be shared.

The behavioural results are neither surprising nor strictly novel, as they are consistent with numerous results obtained with 2D tracking or in immobilised animals (e.g. the optomotor response, Kasumyan & Pavlov 2023 *J Ichthyol*, <https://doi.org/10.1134/S0032945223070081>). But they nicely serve to illustrate the potential of the methodology. Most parts of the method have been published before (as discussed and cited by the authors), but their combination has not. Rather, existing solutions are limited to two-dimensional swimming and tracking, to immobilised fish, or both. By simultaneously tracking body position, posture and eye movements, this new procedure may obviate the need to hold such variables constant, helping observe less artificially constrained behaviour. It is not without its own limitations, though, and can at present only be used for two fish, instead of larger shoals/schools.

The social behaviour of aquatic animals is studied by an active and growing scientific community, and the role of vision in shoaling/schooling is well established (despite some interspecies differences in what other sensory modalities are involved). While designed for and tested on goldfish, the method might conceivably be adapted to a variety of model organisms ranging from medaka to zebrafish, reaching a potentially large audience. As the behavioural results serve more as a proof of principle than as a substantial contribution in its own right, they are less likely to be relevant to the otherwise large community.

Major points

Major point 1: availability of data and code

Lines 353ff: I welcome the authors' willingness to share their code "after publication" (which I assume to mean "immediately upon publication"). This is essential to ensure re-usability in the community. Shared code is sometimes limited to mere parts of data analysis (generating figures from previously recorded tracking data). But given the manuscript's methodological focus, I assume the authors intend to share code that enables readers to replicate the tracking itself.

At present, though, no code has been made available in any form: It was not included among the submitted files available to the reviewers, and the public DOI provided (<https://doi.org/10.6084/m9.figshare.22580230>) currently leads nowhere. Nor does the manuscript contain any description of which kind of code and data are to be shared. It is thus impossible to assess

the scope, quality and re-usability of the code or its documentation.

Communications Biology does not require a formal peer review of code, nor do I consider a proper code review necessary for publication of this manuscript. But without any(!) further information on what is to be shared, and what it will enable readers to do, I cannot assess the relevance of the present manuscript to the community. A mere report on a methodological proof of principle without all the tools needed for replication would be of rather limited interest.

I would kindly ask the authors to provide a detailed description of the code and data to be shared. (Current phrases such as "All data supporting this study's findings" on line 350 are ambiguous.) This could take the shape of a table or plain text, and much of it could be included in the supplementary material. Ideally, they could also provide the code itself to reviewers, but this would not obviate the need for an informative description. Such a description should make it obvious what readers can or cannot do with it, and how it relates to the data being shared (e.g., whether or not pre-trained weights for the convolutional neural networks are included).

If shared code isn't sufficient to enable readers to replicate the authors' method – because, for instance, steps along the pipeline are linked by manual processing steps – this should be clearly stated. In such a case, it might be more helpful to the community to publish the authors' work in the form of a protocols paper (e.g., Nature Protocols?), rather than as a scientific report.

Major point 2: accuracy and reliability of eye tracking

As the authors aim to establish a method for tracking the eye and body movement of goldfish in 3D that is less invasive and less biased than established methods, as well as accurate enough to infer retinal views, a quantitative assessment of its variability and potential biases is crucial. From the present manuscript, I find it hard to assess this quality, but further clarifications will surely help. (See Minor Points 4-6 for specific concerns regarding benchmarking.)

Lines 119-120: The authors report the "average error" of view angle detection as 0 +/- 5 degrees. From the pooled distributions in e.g. Fig. 2b, it is apparent that large deviations of up to +/- 5 degrees are just as common as those near 0 degrees. Thank you for having shared the full distributions, as mere averages of the signed(!) error could have been misleading. If I understand this correctly, the variability given (+/- 5 degrees, +/- 3 pixels) does not refer to a specific mathematical measure such as variance or standard deviation, but to the visual inspection of the width of the nearly uniform-looking distribution? Please indicate.

I'm unsure how to interpret this deviation in terms of tracking quality. It is pooled across all results obtained from a uniform distribution of artificial image parameters. But one would expect certain images (e.g., with a pupil centred on the sclera, directly looking at the camera; cf. Minor Point 5) to be consistently less error-prone than others. Distributions in Fig. 2 appear almost uniform and may hint at persistent differences in tracking quality between test images. How variable is the deviation in view angle detection across different combinations of "true" view angle and camera angle/tilt? Are some combinations more error-prone than others? Do deviations average out over time for any given combination of view and camera angle, or are there systematic biases (e.g., are very eccentric angles consistently over- or underestimated)?

Minor points

Minor point 1: abstract

The abstract adequately summarises the content of the manuscript, though I would recommend minor clarifications: First, please state that your composite method combines previously published methods. The authors are very clear about this elsewhere, but may have omitted it here due to spatial constraints. To compensate, authors may want to consider cutting the rather generic first and last sentences of the abstract (lines 1-2, 23-24). Second, please unambiguously mention "two camera angles" instead of "multiple camera angles" (line 18), and mention that exactly two fish (not more) are tracked in the present study. The same applies elsewhere: The number of "fish in groups" (lines 61) is only clarified later (line 73) as "two".

Minor point 2: potential controls and applications

Lines 142-153, 159-160: The authors correlate the eye movements and relative position of the fish, and reasonably conclude that a causal relationship exists between active vision and shoaling/schooling behaviour. This is consistent with the literature insofar as visual perception is seen as a strong contributing factor, even though other sensory modalities such as the lateral line organ are also implicated (e.g., Partridge & Bitcher 1980 *J Comp Physiol A*, <https://doi.org/10.1007/BF00657647>; Scott et al. 2023 *R Soc Open Sci*, <http://doi.org/10.1098/rsos.221478>). But a key strength of the novel pipeline would be to open up new ways to probe this relationship.

Would the method presented here still work under infrared light (i.e., conditions in which the fish are functionally blind; Chaput et al. 2023 *Behav Brain Res*, <https://doi.org/10.1016/j.bbr.2022.114228>)? Could this be used to disentangle causal and correlational relationships between eye position and following behaviour (e.g., eye positions being putatively informed by input from the lateral line, even in the absence of visual input)? No such control experiments are necessary for the present study, but it would be interesting to highlight the range and limitations of potential applications of the method.

Minor point 3: apparent lack of video data

Supplementary Video 1: The only video submitted with the manuscript is a very nice animated abstract of the manuscript. I was surprised that there is no further video showing the actual pose estimation, tracking of key points via DeepLabCut, eye tracking etc. Such videos might be included among the unspecified data the authors intend to share later (cf. lines 350ff), but this isn't currently apparent. Either way, such videos would be highly relevant to readers. Would you be willing to include them as supplementary videos?

Minor point 4: artificial ground truth vs. manual labelling

Lines 100ff: Precision and robustness of tracking are essential to evaluate the method (cf. Major Point 2). In principle, these can be assessed by comparing a detected variable (e.g. the position of the pupil within the sclera) to its true value. But the authors argue that this ground truth cannot be known for any of their real behavioural recordings, and thus benchmark their method by generating and "tracking" an artificially created set of images. This approach is both creative and pragmatic, but puts a lot of importance on the artificial images chosen. If these are in any way biased, the benchmark no longer captures the actual quality of the method.

Why can the ground truth not be known (or estimated) from actual, unaltered video data? The authors routinely rely on the manual labelling of features by human observers in other parts of their method, e.g. labelling key points along the body to train the convolutional neural networks used for pose estimation. Why can't the benchmarking be conducted on a selection of unaltered video frames for which the "true" eye position has been labelled by an observer?

Minor point 5: ground truth & threshold-based particle detection

Lines 104-113, Figure 2a: The images generated for benchmarking consist of actual still frames of a fish body, overlaid by a smaller, square-shaped drawing of an eye. This drawing combines a bright circle representing the sclera with a randomly placed darker circle representing the pupil. It is blurred (Gaussian smoothed) and tilted to match the camera perspective. These images then undergo threshold-based particle detection. Such algorithms are simple and often robust, but can be vulnerable to image noise: Noisy pixels near the edge of the pupil might cause it to temporarily merge with other particles (the edge of the eye, or dust in the water), biasing the tracked position in the actual video data, but not in the artificial "ground truth" benchmark.

Why does the artificial benchmarking data not contain image noise near the eye? Does this risk overestimating tracking quality? If available, please share evidence that the kind of noise encountered in actual video recordings does not negatively affect tracking. Instead of drawings, could you overlay cropped images of actual fish eyes (with noise) obtained from your own videos? Would these result in the same, or worse, performance?

Minor point 6: ground truth & camera angles

Lines 108-110: In the "ground truth" images, the dark pupil is always surrounded by a portion of the bright area representing the sclera: for a sclera of width w , it is never closer than $3w/8 - (1w/6 + 1w/40) = 0.183w$ to the edge. Gaussian blur will reduce contrast between the pupil and the edge of the sclera but, without noise, should not affect the consistent presence of a brighter zone clearly separating them. Subsequent tilting may further reduce, but not eliminate, this bright zone. This may facilitate threshold-based particle detection here (but not in real data), overestimating tracking quality. Are drawings with uninterrupted bright outlines representative of eyes in actual video data, e.g. of fish swimming towards or away from the camera? This may be a naïve question, but I have not worked with this model organism myself.

Minor point 7: behavioural relevance of tracking precision

Lines 177-178: The authors state that "precision of our eye tracking is high enough for biologically meaningful analyses". Many established methods for eye tracking in 2D, and certainly in immobilised fish, are more accurate. But the authors have chosen a much more challenging problem, of course, and may consider deviations of up to 5 degrees a significant improvement on existing methods for 3D tracking in freely moving animals.

Please (briefly) discuss tracking quality relative to specific phenomena that could or could not be resolved. What is the common range of visual angles under which the leader appears on the retina of the follower? How does that relate to the tracking precision? In a more crowded, multi-fish shoal/school, what are the relevant angles that would have to be distinguished to assess which other fish are actually looked at? Could the method be extended to such applications?

Minor point 8: statistical methods

Figure 5: Bootstrapping is mentioned in the caption of this figure, but not explained anywhere. Please add a description of all statistical methods used to the Methods section.

Minor point 9: transferability to other species

The goldfish is a significant model system in its own right, and novel tools to study it are very welcome. Given the comparable morphological features of many other fish species, it seems quite possible that the method presented here could be adapted to those as well, providing extra value to a much larger scientific community. Doing so is certainly beyond the scope of this work. But it would be highly informative to include an explicit summary of steps that others would need to take in order to adapt the method (and code) to such other species. Or, if adapting the method to other species would be difficult,

the authors should discuss that.

Minor point 10: asymmetry

Lines 138-141: The number of cases in which the leader was in the front-left of the follower (2,524) is much lower than those in the front-right (9,449). How do you explain this? Also, is there an individual preference/bias in each fish, or does this ratio hold across all fish? Has such an asymmetry been reported before?

Minor point 11: literature summary

Lines 36 to 43 summarise the literature, but mix rather different types of eye movement control (from gaze stabilisation to saccades, and from environmental scanning to prey fixation) across multiple species. A more focused literature summary (e.g., with a focus on those aspects of eye movement control most relevant to social interactions in various species) may be more helpful.

Minor point 12: introduction vs. results vs. methods

Lines 72-81: The introduction apparently includes results obtained in the present study. It also contains several citations that refer to previously published methods only, not to previous publications containing the results mentioned. To avoid ambiguity as to what is or isn't novel, this should be disentangled, and all (novel or newly replicated) results moved to that section. Similar concerns apply to technical descriptions, which are currently spread between the Results and Methods section. A more self-contained description would be easier to follow.

Minor point 13: citations

Citations could be placed less ambiguously. For instance, lines 33ff currently read: "Previous studies have attempted (...), yet these studies often overlooked the impact of eye movements [4,5]." The first part of this sentence summarises the content of the literature cited, whereas the final part provides the authors' own judgment on what should have been included in this literature ("these studies overlooked..."). That judgment is professional and entirely defensible, but the citations could be misconstrued as a reference to existing publications agreeing with this judgment. Consider instead: "(...) visual information transfer [4,5], yet these studies (...)"

Throughout the manuscript, consider adjusting the phrasing to match the citation style (small numbers in superscript). "As outlined in [26], (...)" (line 67) could be replaced by "As outlined in our previous publication [26], (...)". The bibliography should contain DOIs where available.

Minor point 14: terminology

Lines 21-22, 76-77, 153 and others: The authors repeatedly refer to "negatively synchronized eye movements" to describe the fact that if one eye tracks the leader, the contralateral eye moves along, even though the leader may not be within its own visual field. I may be unfamiliar with the terminology of the sub-field, but would more commonly refer to this as "conjugate eye movements". This might also avoid confusion with other (entirely correct) statements, e.g. when the authors refer to independent movement of the two eyes enabling the tracking of two separate targets as the "asynchronization of the fish's two eyes" (line 190).

Lines 23, 188, 194 and others: The authors use the term "visual attention" as shorthand for the target of a gaze. Among researchers studying vision in social interactions, equating these concepts appears common and harmless, so I have no objections. However, in neighbouring fields such as neuroscience of vision, "visual attention" often connotes a modulation of neural activity for the preferential processing of some information over others, regardless of gaze (as at least some species can also focus their attention on objects in the periphery). To ensure accessibility to a wider range of readers, it may help to (briefly) clarify the intended meaning.

Minor point 15: software licences

At least some of the software used (e.g. DeepLabCut) is available for free, and the authors apparently intend to freely share (some?) of their own code as well. This is to be highly commended as a valuable contribution to the community. To further facilitate replication and reuse, please name specific licences where appropriate (e.g., Creative Commons CC BY-NC-SA 4.0 or similar), and, if applicable, point out any required software that may be proprietary.

Minor point 16: placeholders

Please make sure that the text of the manuscript corresponds to its intended appearance at publication. At present, it contains (useful, but misplaced) communications directed at the reviewers/editors such as "data supporting this study's findings have been privately uploaded on figshare (...) and will be made public after publication" (line 350f). These should be replaced with their final form now (e.g. "data (...) are publicly available at (...)").

Minor point 17: animal welfare and reporting

The manuscript currently lacks any statement on animal use (e.g., permits obtained regarding the fish facility, animal

husbandry and, where applicable, animal experiments). As per the Nature Portfolio reporting guidelines (<https://www.nature.com/nature-portfolio/editorial-policies/ethics-and-biosecurity#animal-research>), authors may want to consider providing the number of animals used, and naming the relevant laws and regulations as well the authority from which permits were obtained. From the acknowledgments, it appears that a Konstanz fish facility was involved and local regulations may apply, but this is never stated in the main text.

Minor point 18: author contributions

The current author contributions are quite informative, but the authors may want to add further categories from common taxonomies ("software", "data validation", "funding acquisition", etc.).

Minor point 19: supplementary material

Supplementary figures are rather poorly captioned. Some merely feature a title, without any verbal explanation of what is seen in the figure itself. The main body of the article references the supplementary figures, but does not provide such explanations, either.

Closing remarks

Overall, I would like to thank the authors for a stimulating read – and for striving to develop a method that opens up (more) naturalistic behaviour to (more) accurate tracking and analysis. Many existing methods unintentionally limit the range of observable behaviours and thereby systematically bias our empirical data. The challenge now lies in making promising methods like the one presented here genuinely and reliably usable by the community at large, without introducing new biases along the way.

Reviewer #2

(Remarks to the Author)

This manuscript describes a new technique that allows the researchers to track eye movements in freely swimming fish. They use machine learning techniques that they developed previously to track the body in 3D, then locate and track the pupil on the eye from those images.

I have two major criticisms.

1. My primary criticism is that it is very difficult to evaluate how well the algorithm works, because the results are presented in pixels (Fig. 2 and Fig. 5). They should be presented in functionally relevant units. For example, the eye error (Fig. 2g) seems to be ± 3 pixels, but what is a pixel in this context, and how much does that error affect the retinal view reconstruction, as shown in Fig. 5? Also, Fig. 2f seems to indicate that the pupil error is uniformly 0 pixels; is this correct?
2. My secondary criticism regards the use of synthetic data. I understand and support the use of synthetic data to evaluate their algorithm, but I would also like to see the algorithm evaluated against real images. This would require the eye to be tracked manually in some number of images, but I think this is an important step to validate the tracking technique. It would also be helpful to have some metric of similarity between the synthetic images and real images.

Minor comments

1. Ln. 74. Please give a brief explanation of the flow tank methodology, along with referring to reference 27, so that readers do not have to look up the other reference to understand the overall technique.
2. Lns 89-95 and throughout. DeepLabCut, Mask-RCNN, and LSTM. Please explain briefly the different deep learning techniques and justify the use of the different techniques for the different steps.
3. Fig. 2b and throughout. Please define the acronym PDF at some point in the text.
4. Fig. 4a. What do panels iii and iv show? What are the scales of the x and y axes? Are the blobs densities, or different points overlaid on one another with transparency?
5. Fig. 4b. Similarly, is the blob some sort of density or many points overlaid with transparency? If they are points, what does a single point represent? A frame? Are the schematics at the top and bottom of the y axis meant to represent a side view of the fish?
6. Ln. 181. What would be required to expand the current algorithm to multiple cameras? How does the complexity of the algorithm scale with many cameras? Does it become computationally harder or easier with many camera views? Please discuss.
7. Ln. 188. Similarly, please discuss the feasibility and computational complexity of extending the algorithm to many neighbors.
8. SI Table 1. What are the units here? How are the x and y axes defined?

Version 1:

Reviewer comments:

Reviewer #1

(Remarks to the Author)

I'd like to thank the authors for a significant revision of their original submission, which once again proved to be a stimulating read.

Major points:

In my original review, I raised major concerns regarding the availability and documentation of code to replicate the authors' pipeline ("R1.Q1" to "R1.Q4" in the rebuttal letter), and others regarding the accuracy and reliability of eye tracking ("R1.Q5" to "R1.Q7"). The authors have addressed all of these points. This includes significant changes to their previous benchmarking procedure which alleviate some of my concerns, and make the remaining limitations (such as apparent differences between manual labelling and automatic tracking) more transparent.

The authors have since clarified which code will be published ("R1.A1", "R1.A3") and confirmed important matters such as the absence of manual processing steps ("R1.A4"). They have also provided the reviewers access to the code and data intended for publication, including demo videos ("R1.A1"). By journal policy, I refrain from a formal review of the code, but I'm now reasonably confident that the material provided will enable readers to assess the proposed analysis pipeline.

The authors also state that they have "prepared a detailed guideline along with our code on Figureshare, which will be publicly accessible once the paper is published" ("R1.A2"). This is a very welcome addition, and likely crucial to allow replication. However, I could not find said guideline among the figshare files made available. Did I simply overlook it? If not, please add it before publication.

The revised version of the manuscript also makes explicit that readers wishing to use the method in conjunction with an "input scene, fish species, and view angle" that differs from the ones used by the authors will, among other things, have to retrain the underlying networks ("R1.A2", "R1.A19"; lines 222-225 of the revised manuscript; lines 23-26 of the revised Supplementary Information file). This requires considerable time and effort by readers, but should be feasible for labs with the appropriate expertise.

Minor points:

The authors also took great care to address many of the minor points I raised. I would like to highlight a few that I find particularly compelling.

With respect to the authors' response regarding image noise ("R1.A12"), I was surprised that they added (only) white dots. The greater risk to the accurate detection of the (dark) pupils would seem to come from equally dark noisy pixels, rather than from white ones. But I am confident that the information provided by the authors should enable readers to come to their own conclusions, and I thus see no reason to object to publication of the manuscript on this basis.

I do appreciate the inclusion of real still images as input to the benchmarking procedure, as well as their manual labelling ("R1.A12" and "R1.A13"). I'm aware that the latter involved a considerable amount of work, and would like to thank the authors for it. I find it encouraging to see that the analysis pipeline still appears robust, even after application of this updated benchmarking procedure. It's worth noting that beyond a mere difference in precision, there seems to be a systematic bias of about 2 degrees between manually labelled and automatically detected view angles (cf. Figure 3e of the revised manuscript). This may prove problematic for some applications, but since it is clearly shown in one of the main figures, readers will be able to take it into account. This kind of transparency regarding limitations of the methodology is to be commended.

I also appreciate that the authors managed to update their procedure for generating artificial benchmarking images to include frames without an obvious separation between pupil and sclera ("R1.A16"). While there are unavoidable limits to any video tracking, the authors' revised discussion of such caveats is helpful.

The supplementary analysis of the observed left-right asymmetry in behaviour ("R1.A20") is reassuring. Whatever the true cause of this asymmetry (the authors plausibly propose physical asymmetries of the setup as one such possible cause), it does not appear to reflect any bias inherent to the tracking procedure.

I welcome the authors' principled rejection of my suggestion to alter some terminology ("R1.A23"). They are right that their nomenclature, while a bit more unwieldy, is more cautious than the alternative, and I stand corrected.

I'm particularly grateful that the authors now include detailed information about the use of lab animals ("R1.A28"), even including the specific identifier of their relevant government license. Transparency in animal research is a crucial condition of public trust, but - as with any other collective action problem - this does not necessarily translate into proactive communication by the individuals involved. The authors did go that extra step. Thank you!

General observations:

Some possibly confusing typos are present in the revised manuscript (e.g., "infred" on line 250, which could mean either

"inferred" or, more likely, "infrared"). But these can certainly be corrected in the final copyediting process.

Closing remarks:

The revised manuscript represents a significant improvement. I was initially skeptical whether the method presented was sufficiently reliable, and whether it could realistically be replicated by others in the community. With the code and documentation provided, the changes made to the benchmarking procedure, and the more detailed analysis of benchmarking results, I am now much more optimistic. Not least, the authors transparently acknowledge unavoidable limitations (shared with other methods), and discuss necessary steps to adapt their method to alternative experimental setups and species. Far from diminishing their work, this transparency is commendable. The precision and reliability of the method should be sufficient for many research applications, and at present, there appears to be no alternative for the tracking of eye movements in pairs of freely swimming fish in three dimensions.

To reiterate my original summary, developing a method that opens up (more) naturalistic behaviour to (more) accurate tracking and analysis is indeed a valuable scientific contribution. While the method presented here is not without its own (limited) biases and caveats, it represents a very welcome step in the right direction. Tools such as this may shed new light on the role of vision in social interactions, and I look forward to future quantitative research of this kind.

Reviewer #2

(Remarks to the Author)

The authors have largely addressed our criticisms. We have just a few remaining minor comments.

1. Flow tank. I think the authors misunderstood our comment. We know what a flow tank is. Our question was about the size of the working section and the speed of the flow. The size of the working section is now given, but I still do not see any information about the speed of the flow (only "various flow speeds", ln. 79).
2. Fig. 4 caption. Thank you for clarifying the details of the figure. However, the caption of Figure 4 in the revised manuscript is much shorter, and still unclear in places. Some details shown in this answer are missing in the manuscript, for example, "Each point on the heat map represents the position of the front fish, with the following fish's position fixed at the center." This is a key figure, so please make sure the caption is clear.

Reviewer #1 (Remarks to the Author):

Major:

R1.Q1: availability of data and code

Lines 353ff: I welcome the authors' willingness to share their code "after publication" (which I assume to mean "immediately upon publication"). This is essential to ensure re-usability in the community. Shared code is sometimes limited to mere parts of data analysis (generating figures from previously recorded tracking data). But given the manuscript's methodological focus, I assume the authors intend to share code that enables readers to replicate the tracking itself.

R1.A1: Similar to our previous DeepShapeKit work (<https://github.com/preon7/DeepShapeKit>), we will publish our code, including data for demonstration, once the paper is accepted. In the meantime, for your convenience, we have also included the code and data for review. They can be downloaded from <https://figshare.com/s/351d93408e848809c465> or <https://drive.google.com/file/d/1mF2jqTQG6ojuwV-hV819c6X8XPzzyt5/view?usp=sharing>

R1.Q2: At present, though, no code has been made available in any form: It was not included among the submitted files available to the reviewers, and the public DOI provided (<https://doi.org/10.6084/m9.figshare.22580230>) currently leads nowhere. Nor does the manuscript contain any description of which kind of code and data are to be shared. It is thus impossible to assess the scope, quality and re-usability of the code or its documentation.

R1.A2: We appreciate the reviewer's suggestion to describe the reusability and availability of our code. We have prepared a detailed guideline along with our code on Figshare, which will be publicly accessible once the paper is published. We are aware that the absence of this information might have caused confusion for readers and reviewers. In the revised version, we have included a description of the data and code availability. It reads as follows:

"The public dataset includes the coordinates of fish and their eye movements as a CSV file, which we used for statistical analysis, along with a pair of videos as an example of input data. The public code repository contains the fish 4D reconstruction module (DeepShapeKit) and the fish eye tracking modules, enabling users to obtain data on fish swimming poses and eye movements.

In addition to the reconstruction and tracking code, the trained weights for the neural networks involved (Mask R-CNN and DeepLabCut) are included, allowing users to process our shared videos directly. However, users intending

to apply our code to other fish recordings may need to retrain the networks with data tailored to their specific input scene, fish species, and view angle, as well as calculate the projection matrix for the recording camera.

We also provide a demo script that processes the demo videos to extract eye movement data without requiring any manual intervention.”

R1.Q3: Communications Biology does not require a formal peer review of code, nor do I consider a proper code review necessary for publication of this manuscript. But without any(!) further information on what is to be shared, and what it will enable readers to do, I cannot assess the relevance of the present manuscript to the community. A mere report on a methodological proof of principle without all the tools needed for replication would be of rather limited interest.

I would kindly ask the authors to provide a detailed description of the code and data to be shared. (Current phrases such as "All data supporting this study's findings" on line 350 are ambiguous.) This could take the shape of a table or plain text, and much of it could be included in the supplementary material. Ideally, they could also provide the code itself to reviewers, but this would not obviate the need for an informative description. Such a description should make it obvious what readers can or cannot do with it, and how it relates to the data being shared (e.g., whether or not pre-trained weights for the convolutional neural networks are included).

R1.A3: *We agree with the reviewer that the current description may not be very informative. Presenting a table that lists each part of the shared code along with its functionality is an excellent suggestion, as it will provide readers with a clear overview of how they can reuse our code. We have included a table (also below for your convenience) in the revised supplementary materials, and it now reads as follows:*

“The public dataset includes the coordinates of fish and their eye movements as a CSV file, which we used for statistical analysis, along with a pair of videos as an example of input data. The public code repository contains the fish 4D reconstruction module (DeepShapeKit) and the fish eye tracking modules, enabling users to obtain data on fish swimming poses and eye movements.

In addition to the reconstruction and tracking code, the trained weights for the neural networks involved (Mask R-CNN and DeepLabCut) are included, allowing users to process our shared videos directly. However, users intending to apply our code to other fish recordings may need to retrain the networks with data tailored to their specific input scene, fish species, and view angle, as well as calculate the projection matrix for the recording camera.

We also provide a demo script that processes the demo videos to extract eye movement data without requiring manual intervention.”

Table 5: Shared data

Data	Description
Body posture reconstruction	code for reconstruction use DeepShapeKit
Eye tracking	code for extracting eye movement
Data analysis	code for statistical analysis
Sample videos	a pair of video clips for running the demo
Trained network	trained weights for the Mask-RCNN and DeepLabCut

R1.Q4: If shared code isn't sufficient to enable readers to replicate the authors' method – because, for instance, steps along the pipeline are linked by manual processing steps – this should be clearly stated. In such a case, it might be more helpful to the community to publish the authors' work in the form of a protocols paper (e.g., Nature Protocols?), rather than as a scientific report.

R1.A4:

We understand the Reviewer's concern. We do not have any manual intervention in the process. The shared code is fully capable of tracking eye movements from synchronised video records. We included a demo and detailed instructions that guide the user through each step. Additionally, we have outlined the necessary adjustments for applying our method effectively to their custom data in lines 221-222:

“However, this comes at the cost of training Mask-RCNN and DeepLabCut to extract fish masks and keypoints for each view.”

R1.Q5: accuracy and reliability of eye tracking

As the authors aim to establish a method for tracking the eye and body movement of goldfish in 3D that is less invasive and less biased than established methods, as well as accurate enough to infer retinal views, a quantitative assessment of its variability and potential biases is crucial. From the present manuscript, I find it hard to assess this quality, but further clarifications will surely help. (See Minor Points 4-6 for specific concerns regarding benchmarking.)

R1.A5: *We appreciate the constructive suggestions from the reviewer. In the revised version, we have conducted additional analyses and provided detailed descriptions, including the mean values and standard deviations for view angle error, x-axis error, and y-axis error. These metrics are presented for both manually labeled real fish images before perspective transformation and synthesized data after perspective transformation, which serve as ground truth. We have incorporated these results as per the reviewer's suggestions in the revised manuscript and explained the details in the following responses.*

R1.Q6: Lines 119-120: The authors report the “average error” of view angle detection as 0 +/- 5 degrees. From the pooled distributions in e.g. Fig. 2b, it is apparent that large deviations of up to +/- 5 degrees are just as common as those near 0 degrees. Thank you for having shared the full distributions, as mere averages of the signed(!) error could have been misleading. If I understand this correctly, the variability given (+/- 5 degrees, +/- 3 pixels) does not refer to a specific mathematical measure such as variance or standard deviation, but to the visual inspection of the width of the nearly uniform-looking distribution? Please indicate.

R1.A6: *We appreciate the reviewer for raising this concern. The description of “+/- 5 degrees, +/- 3 pixels” was based on a visual inspection of Fig. 2b (now Fig.3) and referred to the range of values rather than the average error. We apologize for this oversight. Upon revising the evaluation, we also recognize that the absence of quantitative descriptions might lead to unclear interpretations of the tracking quality. To enhance clarity and ensure that the tracking quality is accessible to readers, we have included detailed statistical results such as the mean, variance, and the exact range of values (minimum and maximum) in the revised version. We also attached the tables here in **R1.A17** and **R1.A13** for your convenience.*

R1.Q7: I'm unsure how to interpret this deviation in terms of tracking quality. It is pooled across all results obtained from a uniform distribution of artificial image parameters. But one would expect certain images (e.g., with a pupil centred on the sclera, directly looking at the camera; cf. Minor Point 5) to be consistently less error-prone than others. Distributions in Fig. 2 appear almost uniform and may hint at persistent differences in tracking quality between test images. How variable is the deviation in view angle detection across different combinations of “true” view angle and camera angle/tilt? Are some combinations more error-prone than others? Do deviations average out over time for any given combination of view and camera angle, or are there systematic biases (e.g., are very eccentric angles consistently over- or underestimated)?

R1.A7: *We agree that the position of the pupil can potentially affect tracking quality, especially when it is close to the boundary, which may lead to false*

detections. We appreciate the Reviewer for pointing out this issue. The uniform distribution in Fig. 2 (now Fig. 3) appears from the small range of values and low resolution during the image generation. In the revised version, we expand the moving range of the pupil in artificial data, and use higher resolution in the generation step and scale it down to the real image size same as previous ones. Now the error distribution appears as the shape of normal distribution, with the center part represents the value range we used in the first manuscript.

*To investigate whether there is a difference in tracking performance depending on the pupil's location within the sclera, we analyzed the tracking quality when the pupil is positioned at the edge versus the center of the sclera. The result of this analysis is added to the paragraph "Tracking quality across different view angles" on line 155-164, and also presented in detail in **R1.A17***

Minor:

R1.Q8: abstract

The abstract adequately summarises the content of the manuscript, though I would recommend minor clarifications: First, please state that your composite method combines previously published methods. The authors are very clear about this elsewhere, but may have omitted it here due to spatial constraints. To compensate, authors may want to consider cutting the rather generic first and last sentences of the abstract (lines 1-2, 23-24). Second, please unambiguously mention "two camera angles" instead of "multiple camera angles" (line 18), and mention that exactly two fish (not more) are tracked in the present study. The same applies elsewhere: The number of "fish in groups" (lines 61) is only clarified later (line 73) as "two".

R1.A8:

We appreciate the Reviewer's suggestions regarding the abstracts. We have specified this work is followed with our previous DeepShapKit work, and mentioned our method could work for more cameras and larger group size schooling fish and we tested it with two fish scenario and two camera cases in current study. Now it reads as:

*"Eye tracking has emerged as a key method for understanding how animals process visual information, identifying crucial elements of perception and attention. Traditional fish eye tracking often alters animal behavior due to invasive techniques, while non-invasive methods are limited to either 2D tracking or restricting animals after training. Our study introduces a non-invasive technique for tracking and reconstructing the retinal view of free-swimming fish in a large 3D arena without behavioral training. **Using 3D fish body mesh reconstructed by DeepShapeKit**, this method integrates multiple camera angles, deep learning for 3D fish posture reconstruction, perspective transformation, and eye tracking. We evaluated our approach using **data from two fish in a flow tank, captured from two perpendicular viewpoints**, and*

validated its accuracy using human-labeled and synthesized ground truth. Further analysis of eye movements and retinal view reconstruction within leader-follower schooling behavior reveals that fish exhibit conjugate eye movements and focus on neighbors centered in the retinal view. These findings are consistent with previous studies on schooling fish, indirectly validating our method as well. This method offers new insights into animal attention in naturalistic settings and potentially has broader implications for studying collective behavior and advancing swarm robotics.”

R1.Q9: potential controls and applications

Lines 142-153, 159-160: The authors correlate the eye movements and relative position of the fish, and reasonably conclude that a causal relationship exists between active vision and shoaling/schooling behaviour. This is consistent with the literature insofar as visual perception is seen as a strong contributing factor, even though other sensory modalities such as the lateral line organ are also implicated (e.g., Partridge & Bitcher 1980 J Comp Physiol A, <https://doi.org/10.1007/BF00657647>; Scott et al. 2023 R Soc Open Sci, <http://doi.org/10.1098/rsos.221478>). But a key strength of the novel pipeline would be to open up new ways to probe this relationship.

***R1.A9:** We agree with the reviewer that the key strength of this novel pipeline lies in its potential to explore new ways of understanding this relationship. For example, by reconstructing the retina view of 3D free-swimming schooling fish, we can gain deeper insights into why the correlations occur and identify the key features that animals use to sense their neighbors and coordinate their movements. However, before applying the novel pipeline, we need to ensure its effectiveness, which is the primary contribution of this paper. After verification, we will definitely explore these mechanisms in future work. In the revised manuscript, we have updated the discussion to reflect these points on lines 247-253, and it now reads as follows:*

“Building upon the analyses mentioned above, natural extensions for future studies include investigating fish attention in low-light conditions, such as under infrared environment investigating how fish dynamically focus on multiple neighbors, and how their eyes on both sides move synchronously or asynchronously to gather information. Mapping visual input to movement decisions could also shed light on how animals process such collected visual information. Additionally, our methods could potentially be applied to swarm robotics featuring bio-inspired active visual sensing, particularly in interactions with living systems.”

R1Q10: Would the method presented here still work under infrared light (i.e., conditions in which the fish are functionally blind; Chaput et al. 2023 Behav Brain Res, <https://doi.org/10.1016/j.bbr.2022.114228>)? Could this be used to

disentangle causal and correlational relationships between eye position and following behaviour (e.g., eye positions being putatively informed by input from the lateral line, even in the absence of visual input)? No such control experiments are necessary for the present study, but it would be interesting to highlight the range and limitations of potential applications of the method.

R1.A10: We appreciate the reviewer's excellent suggestion for future work. Conducting an experiment with a blind fish could indeed provide valuable insights into the correlation between eye focusing and following behavior. Our pipeline functions as long as the eye is captured by one of the cameras. We reviewed additional fish swimming data recorded with infrared cameras and have provided several examples below. The eyes should be detectable.

Snapshots of a goldfish's eye recorded under infrared light.

R1.Q11: apparent lack of video data

Supplementary Video 1: The only video submitted with the manuscript is a very nice animated abstract of the manuscript. I was surprised that there is no further video showing the actual pose estimation, tracking of key points via DeepLabCut, eye tracking etc. Such videos might be included among the unspecified data the authors intend to share later (cf. lines 350ff), but this isn't currently apparent. Either way, such videos would be highly relevant to readers. Would you be willing to include them as supplementary videos?

R1.A11: We acknowledge the reviewer's constructive suggestion. We have added supplementary video data in our figureshare dataset, <https://figshare.com/s/351d93408e848809c465>.

R1.Q12: artificial ground truth vs. manual labelling

Lines 100ff: Precision and robustness of tracking are essential to evaluate the method (cf. Major Point 2). In principle, these can be assessed by comparing a detected variable (e.g. the position of the pupil within the sclera) to its true value. But the authors argue that this ground truth cannot be known for any of their real behavioural recordings, and thus benchmark their method by

generating and “tracking” an artificially created set of images. This approach is both creative and pragmatic, but puts a lot of importance on the artificial images chosen. If these are in any way biased, the benchmark no longer captures the actual quality of the method.

***R1.A12:** We understand the Reviewer’s concern regarding the use of artificial data for evaluation. If the generated data does not encompass the full range of real-world scenarios, the results may indeed be misleading.*

To address this, we made two updates in our evaluation:

- 1. Add pixel noise on the eye image: randomly place white dots on the image to simulate the particles in the water environment. Increase the moving range of pupil to $(w/2 \pm w/6)$, which covers more diverse cases including pupil is on the sclera border.*
- 2. Manually label eye positions after perspective correction on randomly sampled fish images as real image input. We sampled 110 fish images from the videos and transformed them according to the fish's pose. We then marked the centers and sizes of the sclera and pupil on the fish images using circular markers. After that, we compare the performance of the real images and artificial images.*

*All these results are presented in the next question **R1.A13**. It shows that our pipeline is effective and robust.*

R1.Q13: Why can the ground truth not be known (or estimated) from actual, unaltered video data? The authors routinely rely on the manual labelling of features by human observers in other parts of their method, e.g. labelling key points along the body to train the convolutional neural networks used for pose estimation. Why can’t the benchmarking be conducted on a selection of unaltered video frames for which the “true” eye position has been labelled by an observer?

***R1.A13:** The difficulty in human labelling from the original image is due to the 3D orientation of the real fish, which will distort the distance labelling of the eye position. In order to add human labelling as a comparison, we labelled on the images after perspective correction, where the real fish eye are perpendicular to the camera view. We have labeled approximately 100 frames given the manual effort required.*

A comparison of the human labeling and artificial ground truth is given in the revised manuscript, as well as summarised in the table below:

(a) The pipeline of manually labeling eye positions after perspective transformation and conducting eye tracking as a comparison. Probability distribution function (PDF) of (b, c, d): ground truth and detected eye view angle (b), pupil position (c), and eye position (d). (e, f, g): deviations between the ground truth and eye detection outputs. (h) The pipeline of generating synthesized data as ground truth and conducting eye tracking for a comparison. The PDF of (i, j, k) ground truth and detected eye view angle (i), pupil position (j), and eye position (k). (l, m, n), deviations between the ground truth and eye detection outputs. The deviations are scaled based on the average eye diameter (6.75 mm).

Human labelled ground truth has higher precision comparing to the artificial generated data due to less noise in the video recording since we tried to cover a border range of noise in our artificial data.

Although human errors in manual labeling are inevitable, the evaluation results using manually labeled video frames are consistent with those obtained from artificial data. We have added a section in the revised version discussing the evaluation of real fish images using manually labeled positions.

	Manual Label			Synthesized		
	Angle	Pupil	Eye sclera	Angle	Pupil	Eye sclera
View angle mean error	-2.45°	-	-	-0.10°	-	-
View angle standard deviation	2.65°	-	-	2.95°	-	-
x axis mean error	-	-0.07 mm	0.04 mm	-	-0.09 mm	-0.02 mm
x axis standard deviation	-	0.12 mm	0.12 mm	-	0.33 mm	0.63 mm
y axis mean error	-	0.03 mm	-0.15 mm	-	-0.23 mm	-0.85 mm
y axis standard deviation	-	0.26 mm	0.57 mm	-	0.13 mm	1.19 mm

R1.Q14: ground truth & threshold-based particle detection

Lines 104-113, Figure 2a: The images generated for benchmarking consist of actual still frames of a fish body, overlaid by a smaller, square-shaped drawing of an eye. This drawing combines a bright circle representing the sclera with a randomly placed darker circle representing the pupil. It is blurred (Gaussian smoothed) and tilted to match the camera perspective. These images then undergo threshold-based particle detection. Such algorithms are simple and often robust, but can be vulnerable to image noise: Noisy pixels near the edge of the pupil might cause it to temporarily merge with other particles (the edge of the eye, or dust in the water), biasing the tracked position in the actual video data, but not in the artificial “ground truth” benchmark.

R1.A14: We consider this simple method due to mostly fish eye would not be close to the edge and tested with effectiveness with some noise, but in principle, we could also apply deep learning based methods, such as mask rcnn, to solve the issues. We mentioned this point in the discussion in the revised version:

“In our synthesized validation data, we simulate conditions similar to those observed in our fish recordings, including variations in illumination, shape, slight tilting, and underwater noise such as small particles. However, these simulations may not fully capture all potential noise or variations in eye anatomy that could appear in other video recordings. Through comparison with human-labeled data, we demonstrated that our pipeline is effective for the videos we collected. Nevertheless, we recommend adjusting parameters, such as threshold values in the blob detection step, when adapting our method to different datasets. In cases where the fish is frequently occluded, adding more cameras can help maintain a consistent view of the eye.”

R1Q15: Why does the artificial benchmarking data not contain image noise near the eye? Does this risk overestimating tracking quality? If available, please share evidence that the kind of noise encountered in actual video recordings does not negatively affect tracking. Instead of drawings, could you overlay cropped images of actual fish eyes (with noise) obtained from your own videos? Would these result in the same, or worse, performance?

R1.A15: In most cases, the fish eye has minimal noise, and the white area is clear, allowing us to use a simple background subtraction method. We tested this approach by comparing it to human-labeled real fish image tracking, and our pipeline was effective. While we agree that this final step works well for our dataset, it may not be suitable for other custom datasets. In such cases, we recommend using a deep learning-based tracking method, which should be straightforward to implement.

R1.Q16: ground truth & camera angles

Lines 108-110: In the “ground truth” images, the dark pupil is always surrounded by a portion of the bright area representing the sclera: for a sclera of width w , it is never closer than $3w/8 - (1w/6 + 1w/40) = 0.183w$ to the edge. Gaussian blur will reduce contrast between the pupil and the edge of the sclera but, without noise, should not affect the consistent presence of a brighter zone clearly separating them. Subsequent tilting may further reduce, but not eliminate, this bright zone. This may facilitate threshold-based particle detection here (but not in real data), overestimating tracking quality. Are drawings with uninterrupted bright outlines representative of eyes in actual video data, e.g. of fish swimming towards or away from the camera? This may be a naïve question, but I have not worked with this model organism myself.

R1.A16: We acknowledge the reviewer for raising this critical point. For our pipeline to function, the fish’s eye must be captured by at least one camera, with the pupil and sclera clearly distinguishable. If these two parts cannot be separated due to lighting conditions or the fish’s posture relative to the camera, this algorithm cannot be applied for eye tracking without hardware modifications, such as adding lighting or adjusting the cameras. We have included this note in the revised manuscript.

We understand the Reviewer’s concern regarding the effect of a clear bright zone on the pupil’s detection. In reality, there are instances where the eye area is unevenly illuminated, and the pupil may be close to or even on the dark boundary, leading to low contrast. These variations are present in the manually labeled images.

To address this, we have improved the generation of fish eye images by expanding the movement range of the pupil area to $(w/2 \pm w/6)$. This adjustment ensures that the distance between the pupil border and the eye sclera border can be $((w/3 - w/6) - (w/6 \pm w/20))$, which may be negative, meaning the pupil could be at the border in extreme cases. This enhancement introduces examples where the pupil lacks a bright outline.

We have updated the corresponding paragraph in the revised version to reflect this change. Additionally, when a fish swims towards the camera, the resulting image may lack a bright outline due to resolution limits and underwater noise, which has also been addressed in the discussion section in our updated evaluation.

R1.Q17: behavioural relevance of tracking precision

Lines 177-178: The authors state that “precision of our eye tracking is high enough for biologically meaningful analyses”. Many established methods for eye tracking in 2D, and certainly in immobilised fish, are more accurate. But the

authors have chosen a much more challenging problem, of course, and may consider deviations of up to 5 degrees a significant improvement on existing methods for 3D tracking in freely moving animals.

Please (briefly) discuss tracking quality relative to specific phenomena that could or could not be resolved. What is the common range of visual angles under which the leader appears on the retina of the follower? How does that relate to the tracking precision? In a more crowded, multi-fish shoal/school, what are the relevant angles that would have to be distinguished to assess which other fish are actually looked at? Could the method be extended to such applications?

R1.A17: We appreciate the Reviewer's constructive comments. In our current dataset, we observe that the follower fish tends to keep its eye on the leader when the leader is within approximately 100 degrees in front. This information is added in the revised manuscript at Line 172-176:

"As a result, we obtained a dataset comprising 45,499 data points, divided into 7,860 cases where the leader was in the front-right position relative to the follower ($a > 0$), and 37,639 cases where the leader was in the front-left position ($a < 0$) (see Fig. 4a). Over the leader-follower schooling behavior, the front fish mostly appears within 100° in front."

Since the following fish's eye movement is correlated to the front fish position as shown in Fig. 4b, we can analyze the tracking quality across different pupil positions to show if the leader position will affect the tracking precision.

Given that a goldfish's pupil primarily moves along the x-axis, we defined the edge case based on the distribution of the pupil's x-coordinate as shown in Supplementary Fig. 6 and also enclosed above for your convenience. Specifically, we categorized the pupil's x-coordinate below the first quartile or above the third quartile as the edge case, representing 50% of the data. The center case was defined as the pupil's x-coordinate between the first and third quartiles, accounting for the remaining 50% of the data. A visual description is also provided.

The analysis revealed that the mean and standard deviation of the view angle error were -0.43° and 3.16° for the edge case, and 0.31° and 2.7° for the center case (See Supplementary Tab. 3 and also enclosed below for your convenience). While the error is slightly higher at the edge of the sclera, overall, there is no significant impact on tracking quality. Therefore, the leader's position will not cause precision difference in the eye tracking.

Schematic illustrating data splitting to assess tracking precision based on eye position

When the leader is behind, the tracked fish often redirects its gaze. Since our tracking quality is consistent across various eye view angles (as shown in the revised evaluation), the precision of tracking is not significantly impacted by the leader’s position.

With an increase in the number of fish, there may be cases where two leader fish are close to each other. Our method can distinguish between two fish if they are separated by at least 5 degrees. We have included a discussion of the applicability of our method to different situations in the revised version:

“Additionally, although our current example features two fish swimming in relatively similar directions, our method is versatile enough to be applied to scenarios with freely-swimming fish in large numbers. Based on detection accuracy, we can identify which fish the tracked fish is looking at if there are two leading fish separated by at least 5° in its field of view.”

	View angle		Pupil position	
	Q1 & Q3	Q2	Q1 & Q3	Q2
mean error	-0.43°	0.31°	-0.17 mm	-0.01 mm
standard deviation	3.16°	2.70°	0.30 mm	0.34 mm
minimum error	-9.38°	-5.01°	-1.03 mm	-0.84 mm
maximum error	7.96°	7.06°	0.51 mm	0.90 mm

R1.Q18: statistical methods

Figure 5: Bootstrapping is mentioned in the caption of this figure, but not explained anywhere. Please add a description of all statistical methods used to the Methods section.

R1.A18: We apologize for the missing description. In the bootstrap analysis, we sampled a subset of front fish positions, using 1,000 data points per sample, and repeated this process for 200 iterations to obtain a distribution of the position standard deviations. This detailed description has been added to a new section titled 'Statistical Methods' in the revised version. It reads as:

“we used bootstrap analysis to calculate standard deviation of leader fish's position on x-axis and y-axis. We randomly sample 1000 data points from the result leader fish positions and calculate the standard deviation within the sample for 200 iterations, and plot the 200 standard deviations as a distribution curve. This sampling method is applied to x-axis and y-axis independently.”

R1.Q19: transferability to other species

The goldfish is a significant model system in its own right, and novel tools to study it are very welcome. Given the comparable morphological features of many other fish species, it seems quite possible that the method presented here could be adapted to those as well, providing extra value to a much larger scientific community. Doing so is certainly beyond the scope of this work. But it would be highly informative to include an explicit summary of steps that others would need to take in order to adapt the method (and code) to such other species. Or, if adapting the method to other species would be difficult, the authors should discuss that.

R1.A19: We appreciate the Reviewer's comment. Our goal is to develop a tracking method that can be applied to a range of underwater and some land species. However, due to constraints in time and resources, we have only tested with goldfish. In principle, our method is applicable to any animal with a similar eye structure, provided that key point detection and eye image detection are employed. Adapting our method to other fish species would require training the detection network with species-specific data. To clarify how to adjust our method for different camera setups or target species, we have included a detailed description of the adaptation steps in the discussion section of the revised version:

“The greater the number of cameras, the higher the precision, especially when multiple views capture eye movements. However, this comes at the cost of training Mask-RCNN and DeepLabCut to extract fish masks and keypoints for

each view. The same training process is required when applying our method to other fish species.”

R1.Q20: asymmetry

Lines 138-141: The number of cases in which the leader was in the front-left of the follower (2,524) is much lower than those in the front-right (9,449). How do you explain this? Also, is there an individual preference/bias in each fish, or does this ratio hold across all fish? Has such an asymmetry been reported before?

R1.A20: We appreciate the Reviewer’s comment. Although we applied same filter and parameters for the left and right side formations, the results displayed an asymmetric distribution. This asymmetry may arise because of the asymmetry arena within the flow tank, where one side is transparent and the other is not.

To address the issue of unbalanced data, we applied a bootstrap method to analyze the correlation between fish position and eye movement, which yielded consistent results. Our method and the result plot are:

“In each iteration, we randomly sample 3,000 data points from both the left cases ($x < 0$) and the right cases ($x \geq 0$), resulting in a total of 6,000 points per batch. We calculate the regression function for each batch and repeat this process for 200 iterations. The results show an average slope of -0.00195 and an average intercept of -0.132, with standard deviations of 3.46×10^{-5} and 6.85×10^{-4} , respectively.”

Distribution of the slopes and intercepts of the bootstrapped data.

R1.Q20: literature summary

Lines 36 to 43 summarise the literature, but mix rather different types of eye movement control (from gaze stabilisation to saccades, and from environmental scanning to prey fixation) across multiple species. A more focused literature summary (e.g., with a focus on those aspects of eye

movement control most relevant to social interactions in various species) may be more helpful.

R1.A20: *We appreciate the Reviewer's comment. We agree that a focused literature review can provide readers with a better overview of current studies on eye movement in animal social interactions. We have revised the related works section to emphasize research specifically on eye movements within social contexts. Now it reads as:*

"In nature, eyes are constantly in motion, providing information of the environment [6,7], prey localization [8], and social interaction [9]. Numerous studies have correlated target of gaze with social perception, seeking evidence of vision-based social interactions across various species. For example, mice combine head and eye movements to survey their surroundings to participate in social interactions and visually follow objects [9]. Common marmosets move their eyes to scan the face region to recognize conspecifics [10]. In birds such as starlings, individuals perform lateral scans to gather surrounding information and social cues simultaneously [11]. Goldfish, in particular, exhibit complex and controlled eye movements, characterized by regular saccadic steps and spontaneous side-to-side movements, which are crucial for stabilizing their visual field and gathering environmental information, and engaging in social learning [12-14]."

And also show the new citations here.

Those related works about eye movement in social interaction are:

Daly et.al. 2016 Nature Communications 7, 12140 <https://doi.org/10.1038/ncomms12140>

Land et.al. 2019 Vision Research, Volume 162 <https://doi.org/10.1016/j.visres.2019.06.004>

Ben-Simon et.al. 2009 Journal of Neuroscience Methods, Volume 184 <https://doi.org/10.1016/j.jneumeth.2009.08.006>

Meyer et.al. 2020 Current Biology, Volume 30 <https://doi.org/10.1016/j.cub.2020.04.042>

Kotani et.al. 2017 Psychoneuroendocrinology Volume 83 <https://doi.org/10.1016/j.psyneuen.2017.05.009>

Butler et.al. 2016 Animal Behaviour Volume 121 <https://doi.org/10.1016/j.anbehav.2016.08.002>

Hermann et.al. 1971 Vision Research, Volume 11 [https://doi.org/10.1016/0042-6989\(71\)90243-4](https://doi.org/10.1016/0042-6989(71)90243-4)

Easter et.al. 1971 Vision Research, Volume 11 [https://doi.org/10.1016/0042-6989\(71\)90244-6](https://doi.org/10.1016/0042-6989(71)90244-6)

Blane et.al. 2024 Behavioural Processes Volume 217 <https://doi.org/10.1016/j.beproc.2024.105021>

R1.Q21: introduction vs. results vs. methods

Lines 72-81: The introduction apparently includes results obtained in the present study. It also contains several citations that refer to previously published methods only, not to previous publications containing the results mentioned. To avoid ambiguity as to what is or isn't novel, this should be disentangled, and all (novel or newly replicated) results moved to that section. Similar concerns apply to technical descriptions, which are currently spread between the Results and Methods section. A more self-contained description would be easier to follow.

R1.A21: We apologize for the confusing structure. In the revised version, we describe the entire pipeline: 3D posture tracking, eye tracking, and retinal view reconstruction. For 3D posture tracking, we applied DeepShapeKit, a published method for 4D body mesh reconstruction. The results section primarily focuses on validating our new method for 3D eye tracking. It has now been rephrased in the revised manuscript.

We have kept the details of the method in the Methods section following the main text, as required by the Communications Biology guidelines: <https://www.nature.com/documents/commsj-life-style-formatting-guide-accept.pdf>

R1.Q22: citations

Citations could be placed less ambiguously. For instance, lines 33ff currently read: "Previous studies have attempted (...), yet these studies often overlooked the impact of eye movements [4,5]." The first part of this sentence summarises the content of the literature cited, whereas the final part provides the authors' own judgment on what should have been included in this literature ("these studies overlooked..."). That judgment is professional and entirely defensible, but the citations could be misconstrued as a reference to existing publications agreeing with this judgment. Consider instead: "(...) visual information transfer [4,5], yet these studies (...)"

Throughout the manuscript, consider adjusting the phrasing to match the citation style (small numbers in superscript). "As outlined in [26], (...)" (line 67) could be replaced by "As outlined in our previous publication [26], (...)". The bibliography should contain DOIs where available.

R1.A22: We appreciate the Reviewer's suggestion. Some citations were previously placed in less accurate positions. We have adjusted these citations to follow the mention of the cited work and ensured that the phrasing is consistent throughout the manuscript. These revisions, along with the updated bibliography, are reflected in the revised version.

R1.Q23: terminology

Lines 21-22, 76-77, 153 and others: The authors repeatedly refer to “negatively synchronized eye movements” to describe the fact that if one eye tracks the leader, the contralateral eye moves along, even though the leader may not be within its own visual field. I may be unfamiliar with the terminology of the sub-field, but would more commonly refer to this as “conjugate eye movements”. This might also avoid confusion with other (entirely correct) statements, e.g. when the authors refer to independent movement of the two eyes enabling the tracking of two separate targets as the “asynchronization of the fish’s two eyes” (line 190).

R1.A23: We appreciate the Reviewer pointing out the point.

*Conjugate eye movements often refer to both eyes moving in the same direction simultaneously. It is a coordinated action where the eyes move together to **maintain binocular vision** and **align the visual field**, ensuring that an image stays centered on the retina of both eyes. This type of movement is typical in humans and many animals, including fish when they need to focus on an object or track movement.*

In this paper, the observed negatively synchronized eye movements could suggest that fish are using conjugate eye movements to track their neighbors. However, we do not have strong data to definitively support this conclusion. Therefore, we believe it is more accurate and cautious to describe the phenomenon as negatively synchronized eye movements, reflecting only what we observed.

R1.Q24: Lines 23, 188, 194 and others: The authors use the term “visual attention” as shorthand for the target of a gaze. Among researchers studying vision in social interactions, equating these concepts appears common and harmless, so I have no objections. However, in neighbouring fields such as neuroscience of vision, “visual attention” often connotes a modulation of neural activity for the preferential processing of some information over others, regardless of gaze (as at least some species can also focus their attention on objects in the periphery). To ensure accessibility to a wider range of readers, it may help to (briefly) clarify the intended meaning.

R1.A24: We apologize for the ambiguous description. To reduce potential misunderstandings, we have replaced all instances of 'visual attention' with 'target of gaze' in the revised version. We appreciate the Reviewer's suggestion and explanation.

R1.Q25: software licences

At least some of the software used (e.g. DeepLabCut) is available for free, and the authors apparently intend to freely share (some?) of their own code as well. This is to be highly commended as a valuable contribution to the community. To further facilitate replication and reuse, please name specific licences where appropriate (e.g., Creative Commons CC BY-NC-SA 4.0 or similar), and, if applicable, point out any required software that may be proprietary.

R1.A25: We appreciate the Reviewer's comment. Our tracking method and associated software are published under the CC-BY-4.0 license, and all included software packages are open source. We apologize for the oversight regarding the license declaration. The license information has been added to the code availability section in the revised version:

"All the data analyses were performed using custom scripts written in Python (Python Software Foundation, 2018). Our published codes are licensed under CC-BY-4.0."

R1.Q26: placeholders

Please make sure that the text of the manuscript corresponds to its intended appearance at publication. At present, it contains (useful, but misplaced) communications directed at the reviewers/editors such as "data supporting this study's findings have been privately uploaded on figshare (...) and will be made public after publication" (line 350f). These should be replaced with their final form now (e.g. "data (...) are publicly available at (...)").

R1.A26: We appreciate the Reviewer's comment. We will ensure that the revised manuscript reflects the final published appearance. Text related to the review stage has been updated accordingly in the revised version.

R1.Q27: animal welfare and reporting

The manuscript currently lacks any statement on animal use (e.g., permits obtained regarding the fish facility, animal husbandry and, where applicable, animal experiments). As per the Nature Portfolio reporting guidelines (<https://www.nature.com/nature-portfolio/editorial-policies/ethics-and-biosecurity#animal-research>), authors may want to consider providing the number of animals used, and naming the relevant laws and regulations as well the authority from which permits were obtained. From the acknowledgments, it appears that a Konstanz fish facility was involved and local regulations may apply, but this is never stated in the main text.

R1.A28: We appreciate the Reviewer's comment. We have included details about the number of animals used and the facilities for their care in the 'Data Collection' section, along with the existing regulatory information.

"The fish in each experiment are selected randomly from 32 goldfish kept in the animal care facility at the University of Konstanz. All animal handling and experimental procedures were approved by Regierungspräsidium Freiburg, 35-9185.81/G-17/90."

R1.Q29: author contributions

The current author contributions are quite informative, but the authors may want to add further categories from common taxonomies ("software", "data validation", "funding acquisition", etc.).

R1.A29: We appreciate the Reviewer's comment. Following this helpful suggestion, we have added the names of the authors responsible for the common tasks to the 'Contributions' section in the revised version.

R1.Q30: supplementary material

Supplementary figures are rather poorly captioned. Some merely feature a title, without any verbal explanation of what is seen in the figure itself. The main body of the article references the supplementary figures, but does not provide such explanations, either.

R1.A30: We appreciate the Reviewer's comment and apologize for the oversight regarding the missing captions. After reviewing the supplementary PDF, we noted that many figures lacked explanatory captions. To address this, we have added detailed descriptions to the captions of all supplementary figures in the revised version, ensuring that each figure's content is clear and understandable without needing to refer back to the main text.

Reviewer #2 (Remarks to the Author):

Major:

R2.Q1: My primary criticism is that it is very difficult to evaluate how well the algorithm works, because the results are presented in pixels (Fig. 2 and Fig. 5). They should be presented in functionally relevant units. For example, the eye error (Fig. 2g) seems to be ± 3 pixels, but what is a pixel in this context, and how much does that error affect the retinal view reconstruction, as shown in Fig. 5? Also, Fig. 2f seems to indicate that the pupil error is uniformly 0 pixels; is this correct?

R2.A1: *We appreciate the Reviewer's comment and agree that using a functionally relevant unit improves the clarity of the results. In the revised version, we have converted the pixel unit to millimeters based on fish body length.*

About the Reviewer's confusion:

- 1. Pixels here are referring to the length on the cropped eye image, where the size of the sclera area is around 6.75 mm, corresponding to 9 pixels. According to this, we converted the pixel unit to millimeters and each pixel represents 0.75 mm.*
- 2. The impact of the tracking error in view reconstruction is reflected in the view angle error, which is the angle between reconstructed view angle and the ground truth view angle. Here our eye tracking shows an average error of 0.1° and a standard deviation of 2.65 degrees.*
- 3. The zero pupil error observed previously was due to the precision of the generation process, where the resolution of the generated and detected images was the same, so the detection result is pixel perfect. In the revised version, we addressed this by generating eye images at a higher resolution and then scaling them down to the detection resolution. This allows us to label pupil positions with sub-pixel precision. We have also included additional results in the section evaluating manually labeled eye data.*

R2.Q2: My secondary criticism regards the use of synthetic data. I understand and support the use of synthetic data to evaluate their algorithm, but I would also like to see the algorithm evaluated against real images. This would require the eye to be tracked manually in some number of images, but I think this is an important step to validate the tracking technique. It would also be helpful to have some metric of similarity between the synthetic images and real images.

R2.A2: *We appreciate the Reviewer's comment. Including real fish eye images indeed enhances the persuasiveness of the evaluation results. To test the performance on real fish images, we sampled 110 fish images from the videos and transformed them according to the fish's pose. We then marked the positions and sizes of the sclera and pupil on the fish images manually. By doing so, we get a set of real fish eye images with labelled eye position, which we can put in our eye tracking pipeline and check the tracking result.*

The comparison between real fish eye tracking and synthesized eye image tracking is summarized in R1.A13. In summary, the tracking quality of both real and artificial images is similar.

Minor:

R2.Q3: Ln. 74. Please give a brief explanation of the flow tank methodology, along with referring to reference 27, so that readers do not have to look up the other reference to understand the overall technique.

R2.A3: *We apologize for the missing explanation. As some readers may not be familiar with the flow tank and its use in hydrodynamic experiments, we have added a description to clarify its purpose and operation in the revised version. Also mention the text here:*

It read as:

“It is a device that allows two fish to swim in a tunnel with a constant water flow, enabling us to observe their swimming behavior in a fixed location.”

R2.Q4: Lns 89-95 and throughout. DeepLabCut, Mask-RCNN, and LSTM. Please explain briefly the different deep learning techniques and justify the use of the different techniques for the different steps.

R2.A4: *We apologize for the missing description. While these network structures are common in computer science, they may be confusing to those from other fields. We have provided a brief explanation of their usage and the reasons for choosing them in the revised version. You need to explain what are the the deep learning algorithms for and why. And then put the texts here:*

“we optimize the position and kinematics of this model to minimize the discrepancies between the model and the silhouettes of real fish tracked by Mask-RCNN as well as key points of the body central line tracked by DeepLabCut. An LSTM (long short-term memory network)-based smoother is applied to smooth the 3D meshes in sequence. With the 3D body mesh over the global coordinate, we get fish body posture in 3D.”

R2.Q5: Fig. 2b and throughout. Please define the acronym PDF at some point in the text.

R2.A5: *We thank the Reviewer’s suggestion. We added the description in the Fig. 2b (Now Fig. 3) caption:*

“We plot the probability distribution function (PDF) of each detection result.”

R2.Q6: Fig. 4a. What do panels iii and iv show? What are the scales of the x and y axes? Are the blobs densities, or different points overlaid on one another with transparency?

R2.A6: *We apologize for the missing description. In Fig. 4a, we present the heat map of the front fish's position relative to the following fish. The x and y axes represent spatial positions ranging from -50 cm to 50 cm. We have added*

additional details to the caption of Fig. 4 in the revised version to clarify this. The blobs in the heat map represent data points overlaid with transparency.

R2.Q7: Fig. 4b. Similarly, is the blob some sort of density or many points overlaid with transparency? If they are points, what does a single point represent? A frame? Are the schematics at the top and bottom of the y axis meant to represent a side view of the fish?

***R2.A7:** We appreciate the Reviewer's comment. In Fig. 4b, the blobs represent data points overlaid with transparency, where each data point denotes the relative position of a fish in a frame. The x-axis indicates the relative angle between the two fish from a bottom view, while the y-axis represents the eye position of the following fish from left (top) to right (bottom). The illustration on the y-axis shows the fish eye position from a side view corresponding to these values. We have added detailed descriptions of the x and y axes to the caption in the revised version:*

*“(iii): The front fish position heat map, viewed from the bottom side, shows the relative positions of the front fish from the perspective of the following fish looking upwards. Each point on the heat map represents the position of the front fish, with the following fish's position fixed at the center. The x and y axes span a range of ± 50 cm, capturing the spatial distribution of the front fish.
(iv): The front fish position heat map, viewed from the front side, presents the relative positions of the front fish from the perspective of the following fish facing left. Again, each point represents the front fish's position, with the following fish at the center. The x and y axes cover a range of ± 50 cm, illustrating the positioning patterns during the interaction.
”*

R2.Q8: Ln. 181. What would be required to expand the current algorithm to multiple cameras? How does the complexity of the algorithm scale with many cameras? Does it become computationally harder or easier with many camera views? Please discuss.

***R2.A8:** We appreciate the Reviewer's comment. For the first question, incorporating additional cameras requires new training data from each new view angle for Mask-RCNN and DeepLabCut. Each additional view necessitates extra labeling and training work.*

For the second question: our algorithm is designed to work with input from more than two cameras. The complexity of the algorithm does not change as the input camera amount can be an arbitrary number.

For the final question: while the algorithm itself does not significantly increase computational power requirements since optimization is handled with a matrix

encompassing all view angles, more cameras do involve additional preprocessing.

We have added a discussion on this aspect in the revised version:

“The more cameras there are, the higher the precision, especially when multiple views capture eye movements. However, this comes at the cost of training Mask-RCNN and DeepLabCut to extract fish masks and keypoints for each view. The same training process is required when applying our method to other fish species.”

R2.Q9: Ln. 188. Similarly, please discuss the feasibility and computational complexity of extending the algorithm to many neighbors.

R2.A9: *We appreciate the Reviewer’s comment. Extending the algorithm to handle additional neighboring fish is less complex compared to incorporating multiple cameras. This adjustment involves simply updating the fish count in the code to match the number of fish appearing in the image.*

We have included further details on this in the discussion section of the revised version:

“Additionally, although our current example features two fish swimming in relatively similar directions, our method is versatile enough to be applied to scenarios with freely-swimming fish in large numbers. In general, the more individuals involved, the more challenging it becomes due to frequent overlapping. This would require not only more cameras but also a large, highly precise dataset of fish mask and keypoint labeling for training”

R2.Q10: SI Table 1. What are the units here? How are the x and y axes defined?

R2.A10: *We appreciate the Reviewer’s comment. The standard deviations are measured in millimeters. The x-axis and y-axis directions correspond to the illustration in the main figure 5. We have added these descriptions in the revised version.*

Table 1: Standard deviation of normalized front fish position in meters. The x-axis and y-axis are referring to Figure 5 in the main text.

	Dynamic	Static	Random
x-axis	0.0786 mm	0.0951 mm	8.638 mm
y-axis	0.0251 mm	0.0293 mm	0.862 mm

Reviewer #1 (Remarks to the Author):

R1Q1: I'd like to thank the authors for a significant revision of their original submission, which once again proved to be a stimulating read.

***R1.A1:** Thank you for your constructive comments and the time you've taken to help improve our manuscript. We have added detailed readme file for the figureshare platform.*

Major points:

R1Q2: In my original review, I raised major concerns regarding the availability and documentation of code to replicate the authors' pipeline ("R1.Q1" to "R1.Q4" in the rebuttal letter), and others regarding the accuracy and reliability of eye tracking ("R1.Q5" to "R1.Q7"). The authors have addressed all of these points. This includes significant changes to their previous benchmarking procedure which alleviate some of my concerns, and make the remaining limitations (such as apparent differences between manual labelling and automatic tracking) more transparent.

***R1.A2:** Thanks.*

R1Q3: The authors have since clarified which code will be published ("R1.A1", "R1.A3") and confirmed important matters such as the absence of manual processing steps ("R1.A4"). They have also provided the reviewers access to the code and data intended for publication, including demo videos ("R1.A1"). By journal policy, I refrain from a formal review of the code, but I'm now reasonably confident that the material provided will enable readers to assess the proposed analysis pipeline.

***R1.A3:** Thank you.*

R1Q4: The authors also state that they have "prepared a detailed guideline along with our code on Figureshare, which will be publicly accessible once the paper is published" ("R1.A2"). This is a very welcome addition, and likely crucial to allow replication. However, I could not find said guideline among the figshare files made available. Did I simply overlook it? If not, please add it before publication.

***R1.A4:** Apologies for not including a README file on the Figshare platform initially, as we did for the GitHub repository. We have now added a detailed README to provide clear guidelines.*

R1Q5: The revised version of the manuscript also makes explicit that readers wishing to use the method in conjunction with an "input scene, fish species, and view angle" that differs from the ones used by the authors will, among other things, have to retrain the underlying networks ("R1.A2", "R1.A19"; lines 222-225 of the revised manuscript; lines 23-26 of the revised Supplementary Information file). This requires considerable time and effort by readers, but should be feasible for labs with the appropriate expertise.

***R1.A5:** Currently, we have data for only one species, but we aim to expand our method to be more general by incorporating data from additional species in the future.*

Minor points:

R1Q6: The authors also took great care to address many of the minor points I raised. I would like to highlight a few that I find particularly compelling.

R1.A6: *Thank you.*

R1Q7: With respect to the authors' response regarding image noise ("R1.A12"), I was surprised that they added (only) white dots. The greater risk to the accurate detection of the (dark) pupils would seem to come from equally dark noisy pixels, rather than from white ones. But I am confident that the information provided by the authors should enable readers to come to their own conclusions, and I thus see no reason to object to publication of the manuscript on this basis.

R1.A7: *We appreciate that.*

R1Q8: I do appreciate the inclusion of real still images as input to the benchmarking procedure, as well as their manual labelling ("R1.A12" and "R1.A13"). I'm aware that the latter involved a considerable amount of work, and would like to thank the authors for it. I find it encouraging to see that the analysis pipeline still appears robust, even after application of this updated benchmarking procedure. It's worth noting that beyond a mere difference in precision, there seems to be a systematic bias of about 2 degrees between manually labelled and automatically detected view angles (cf. Figure 3e of the revised manuscript). This may prove problematic for some applications, but since it is clearly shown in one of the main figures, readers will be able to take it into account. This kind of transparency regarding limitations of the methodology is to be commended.

R1.A8: *Thank you.*

R1Q9: I also appreciate that the authors managed to update their procedure for generating artificial benchmarking images to include frames without an obvious separation between pupil and sclera ("R1.A16"). While there are unavoidable limits to any video tracking, the authors' revised discussion of such caveats is helpful.

R1.A9: *Thanks.*

R1Q10: The supplementary analysis of the observed left-right asymmetry in behaviour ("R1.A20") is reassuring. Whatever the true cause of this asymmetry (the authors plausibly propose physical asymmetries of the setup as one such possible cause), it does not appear to reflect any bias inherent to the tracking procedure.

R1.A10: *We agree.*

R1Q11: I welcome the authors' principled rejection of my suggestion to alter some terminology ("R1.A23"). They are right that their nomenclature, while a bit more unwieldy, is more cautious than the alternative, and I stand corrected.

I'm particularly grateful that the authors now include detailed information about the use of lab animals ("R1.A28"), even including the specific identifier of their relevant government license. Transparency in animal research is a crucial condition of public trust, but - as with any other collective action problem - this does not necessarily translate into proactive communication by the individuals involved. The authors did go that extra step. Thank you!

R1.A11: *Thank you for your constructive comments that helped us improve the manuscript.*

General observations:

R1Q12: Some possibly confusing typos are present in the revised manuscript (e.g., "infred" on line 250, which could mean either "inferred" or, more likely, "infrared"). But these can certainly be corrected in the final copyediting process.

R1.A12: *Done*

Closing remarks:

R1Q13: The revised manuscript represents a significant improvement. I was initially skeptical whether the method presented was sufficiently reliable, and whether it could realistically be replicated by others in the community. With the code and documentation provided, the changes made to the benchmarking procedure, and the more detailed analysis of benchmarking results, I am now much more optimistic. Not least, the authors transparently acknowledge unavoidable limitations (shared with other methods), and discuss necessary steps to adapt their method to alternative experimental setups and species. Far from diminishing their work, this transparency is commendable. The precision and reliability of the method should be sufficient for many research applications, and at present, there appears to be no alternative for the tracking of eye movements in pairs of freely swimming fish in three dimensions.

R1.A13: *Thank you*

R1Q14: To reiterate my original summary, developing a method that opens up (more) naturalistic behaviour to (more) accurate tracking and analysis is indeed a valuable scientific contribution. While the method presented here is not without its own (limited) biases and caveats, it represents a very welcome step in the right direction. Tools such as this may shed new light on the role of vision in social interactions, and I look forward to future quantitative research of this kind.

R1.A14: *Thank you. We appreciate this thoughtful feedback.*

Reviewer #2 (Remarks to the Author):

The authors have largely addressed our criticisms. We have just a few remaining minor comments.

R2Q1: 1. Flow tank. I think the authors misunderstood our comment. We know what a flow tank is. Our question was about the size of the working section and the speed of the flow. The size of the working section is now given, but I still do not see any information about the speed of the flow (only "various flow speeds", ln. 79).

R2A1: *We have added the flow information in the revised manuscript. Now it reads as:*

Line 77-80: "We validated our methodology with synchronized videos from two perpendicular views of two goldfish swimming freely in a flow tank at various flow speeds ranging from 1.2 BL/s (Body length per second) to 1.6 BL/s with an interval of 0.1 BL/s (see supplementary Fig. 1 and [26] for details)."

R2Q2: 2. Fig. 4 caption. Thank you for clarifying the details of the figure. However, the caption of Figure 4 in the revised manuscript is much shorter, and still unclear in places. Some details shown in this answer are missing in the manuscript, for example, "Each point on the heat map represents the position of the front fish, with the following

fish's position fixed at the center.” This is a key figure, so please make sure the caption is clear.

R2A2: We apologies for the oversight, we have added details for fugure 4 caption. Now it reads as:

“Eye tracking of two swimming goldfish exhibiting leader-follower behavior. a Selected pairs of goldfish within swimming groups. (i) Two-fish relationship from the bottom view. (ii) Front fish position density map on a 3D sphere. All 45,499 pairs of data are binned in 12 degrees increments, with a radius of 17.5 cm. (iii): The front fish position heat map, viewed from the bottom side, shows the relative positions of the front fish from the perspective of the following fish looking upwards. Each point on the heat map represents the position of the front fish, with the following fish’s position fixed at the center. The x and y axes span a range of ± 50 cm, capturing the spatial distribution of the front fish. (iv): The front fish position heat map, viewed from the front side, presents the relative positions of the front fish from the perspective of the following fish facing left. Again, each point represents the front fish’s position, with the following fish at the center. The x and y axes cover a range of ± 50 cm, illustrating the positioning patterns during the interaction. **b** The correlation between the normalised eye movements ranging from -0.4 (rightmost position) to 0.4 (leftmost position) and relative position angles”